# TEAD-independent mechanisms of YAP function in cardiomyocyte cell cycle reentry

Bing Xie[1,2] , Jeffrey Steimle[1] , Vaibhav Deshmukh[1] , Lin Liu[1], Chang-Ru Tsai[1], Todd R Heallen[4], Wyatt Paltzer[1] , Yuka Morikawa[4] , Fansen Meng[1], Jun Wang[4] , James F Martin[1,2,3,4]

**Adult mammalian hearts exhibit limited regenerative capacity because of the restricted renewal of cardiomyocytes. Recent studies reveal that mammalian hearts exhibit transient regenerative potential within a short time frame after birth, suggesting a regulatory mechanism that prevents adult hearts from initiating a regenerative response to cardiac injury. Here, we discovered that an active form of YAP, named YAP6SA, which is not inhibited by the Hippo signaling pathway and does not interact with TEADs, induces cardiomyocyte cell cycle reentry. In addition, YAP6SA interacts with scaffold protein MPDZ to regulate Rho GTPases and promote cell cycle progression in cardiomyocytes (CMs). Importantly, YAP6SA overexpression is well tolerated in mammalian hearts. These findings provide new insights into YAP function in cardiomyocytes.**

## Introduction

Unlike nonmammalian vertebrates, such as zebrafish, whose hearts can fully regenerate in response to injury throughout life (Poss et al, 2002), mature mammalian hearts harbor terminally differentiated cardiomyocytes (CMs). CM loss during cardiac ischemia is irreversible and induces cardiac fibroblast activation, which mediates scar formation and leads to impaired cardiac contractility (Abbate et al, 2006; Dobaczewski et al, 2010; van Berlo & Molkentin, 2014). However, recent studies have demonstrated that the neonatal mouse heart has transient regenerative potential during the first 7 d of life, suggesting a regulatory mechanism that prevents mature hearts from initiating a regenerative response to ischemic injury (Porrello et al, 2011).

The Hippo signaling pathway (HSP), an evolutionarily conserved anti-growth pathway, was initially identified in *Drosophila* for its regulation of organ and body size (Yu & Guan, 2013). The core HSP components in mammals include the STE20 family protein kinases MST1/2; the LATS1/2 kinases, which are phosphorylated and activated by MST1/2; and YAP, a transcriptional coactivator growth molecule inhibited by LATS1/2. The HSP phosphorylates YAP through LATS1/2, leading to its cytoplasmic retention and degradation, which suppresses its transcriptional activity. When unphosphorylated, active YAP translocates to the nucleus where it interacts with transcription factors, such as TEADs, to drive the expression of genes involved in cell proliferation and differentiation (Meng et al, 2016). Previous studies have revealed that HSP activity is high in postnatal hearts (Wang et al, 2018). We deleted the MST1/2 adapter protein SAV in adult mouse hearts to suppress Hippo signaling activity, and these hearts activate a reparative genetic program after myocardial infarction (Leach et al, 2017). This demonstrates a regulatory role of HSP in cardiac regeneration. Moreover, CM-specific YAP overexpression enhances cardiac function and survival by promoting cardiomyocyte proliferation after myocardial injury (von Gise et al, 2012; Xin et al, 2013; Lin et al, 2014). These findings further clarify how inhibition of the HSP facilitates cardiomyocyte renewal.

YAP lacks intrinsic DNA-binding activity and partners with DNA-binding transcription factors to regulate the expression of target genes (Zhao et al, 2008). Canonically, the TEAD family (TEA domain transcription factors 1–4 in mammals) serves as the primary DNA-binding partner for YAP (Li et al, 2010; Zanconato et al, 2015). TEAD proteins contain a DNA-binding TEA domain but possess a weak transcriptional activation domain (Vassilev et al, 2001). YAP, which contains a strong activation domain, is recruited to TEAD recognition elements in the genome, thereby enabling the transcriptional activation of target genes (Li et al, 2010; Galli et al, 2015). Although the canonical function of YAP in the heart is mediated through its interaction with TEAD transcription factors to drive cell proliferation and survival gene programs, emerging evidence suggests that YAP also exerts important TEAD-independent effects. We developed a CM-specific YAP gain-of-function mouse model named YAP5SA, in which the LATS1/2 phosphorylation sites of the

[1]Department of Integrative Physiology, Baylor College of Medicine, Houston, TX, USA   [2]Genetics and Genomics Graduate Program, Baylor College of Medicine, Houston, TX, USA   [3]Center for Organ Repair and Renewal, Baylor College of Medicine, Houston, TX, USA   [4]Cardiomyocyte Renewal Laboratory, Texas Heart Institute, Houston, TX, USA

Correspondence: jfmartin@bcm.edu
Jeffrey Steimle and Vaibhav Deshmukh's present address is Division of Molecular Cardiovascular Biology, Department of Pediatrics, Cincinnati Children's Hospital Medical Center, University of Cincinnati, Cincinnati, OH, USA

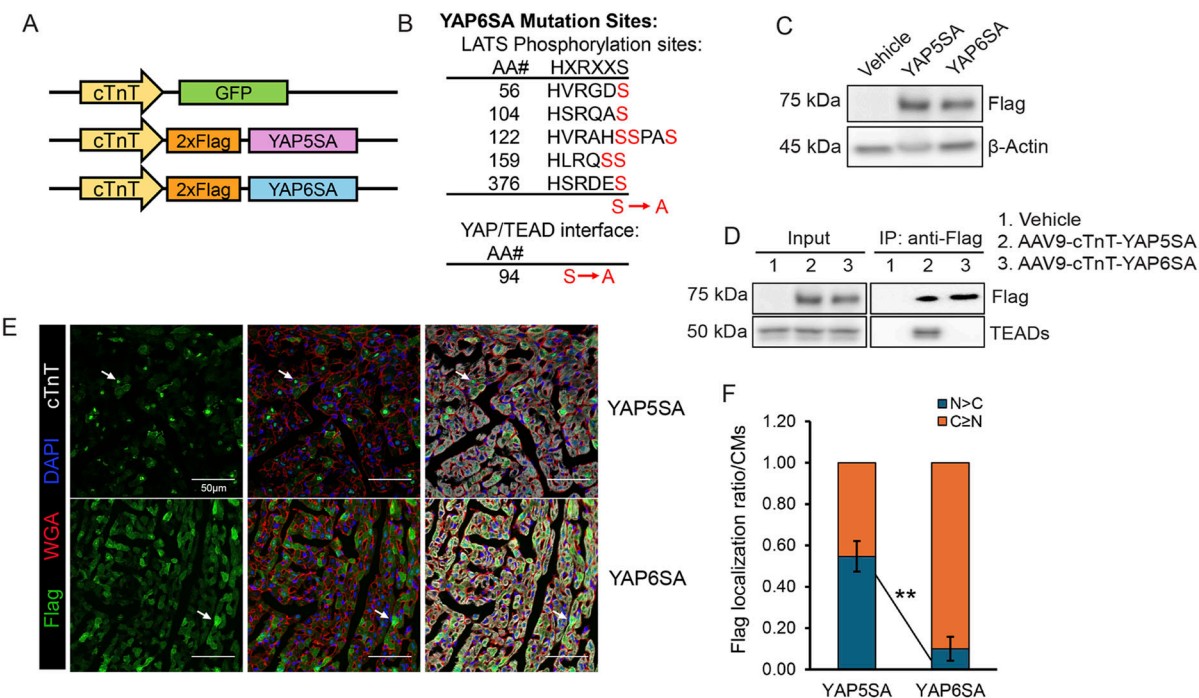

**Figure 1.   Cardiac phenotype of YAP6SA-overexpressing mice.**
**(A)** AAV9-GFP−, AAV9-YAP5SA−, and AAV9-YAP6SA−expressing cassettes. cTnT, cardiac troponin T. **(B)** Mutations of the YAP6SA protein sequence. **(C)** Western blotting results of Flag-YAP5SA and Flag-YAP6SA expression levels after 4 d of AAV9 infection in P6 murine hearts. **(C, D)** Co-IP of YAP5SA and YAP6SA interactors after 4 d of AAV9 infection in P6 murine hearts using the same input/lysates as shown in (C). **(E)** Immunofluorescence of Flag expression in YAP5SA and YAP6SA OE mice 3 d after AAV9 infection in P6 murine hearts. White arrows show the location of nuclear YAP5SA and cytoplasmic YAP6SA in CMs. **(F)** Quantification of the Flag signal in YAP5SA and YAP6SA OE cardiomyocytes (n = 6 each).
Source data are available for this figure.

YAP protein are mutated to completely bypass HSP repression (Monroe et al, 2019). YAP5SA adult mouse hearts exhibit cardiac hyperplasia, marked by a thickened ventricular wall and shortened survival. In addition, YAP5SA alters the expression of multiple genes and enhances chromatin accessibility. We determined that 40% of YAP5SA up-regulated genes contain TEAD-binding motifs, and YAP5SA initiates both positive and negative feedback loops of CM proliferation. Therefore, the mechanism of YAP5SA function is complex and remains to be fully defined. Here, we found that active YAP induces cardiomyocyte cell cycle activity through TEAD-independent mechanisms. To test this, we generated a modified YAP protein expression system named YAP6SA, in which the serine at residue 94 is mutated to alanine, thereby inhibiting the interaction between YAP and TEADs (Li et al, 2010), while still bypassing HSP inhibitory regulation. Our findings reveal a previously unknown flexibility in YAP's mechanism of action, suggesting that cardiac renewal can be achieved through TEAD-independent routes.

## Results

### Generation of a YAP6SA gain-of-function expression cassette

We generated an AAV9 virus expressing YAP6SA tagged with the Flag epitope and transcribed under the cardiac troponin T regulatory element, which directs protein expression in cardiomyocytes (Forough et al, 2011) (Fig 1A). YAP6SA contains six serine-to-alanine mutations at key regulatory sites. Five of these mutations correspond to the LATS1/2 phosphorylation sites found in YAP5SA, whereas the sixth is a unique substitution at residue 94, where serine is replaced by alanine (Fig 1B). We packaged the AAV9 virus with YAP5SA or YAP6SA expression sequences and infected mice at 6 d of age (P6). We performed immunofluorescence (IF) to detect the Flag signal from day 1 to day 7 after AAV9 infection, using the cardiac troponin T (cTnT) as the CM marker. We observed that YAP5SA and YAP6SA expression initiates at day 3 post infection (Fig S1A). Western blotting revealed YAP5SA and YAP6SA expression in murine hearts after AAV9 delivery, and co-immunoprecipitation (co-IP) confirmed the disrupted interaction between TEAD and YAP6SA (Fig 1C and D). We observed that although Flag-YAP5SA and Flag-YAP6SA are specifically and ubiquitously expressed in CMs, YAP6SA exhibited a decreased nuclear localization and an increased ratio in the cytoplasm (Figs 1E and F and S1B). This result suggests that the YAP/TEAD interaction influences the cellular distribution of YAP.

### YAP6SA overexpression is safely tolerated in vivo

To determine the effect of YAP6SA overexpression (OE) on cardiac homeostasis, we first examined mouse survival among AAV9-GFP, AAV9-YAP5SA, and AAV9-YAP6SA OE postnatal groups. We observed that YAP6SA OE mice had a significantly extended survival rate

compared with YAP5SA OE mice and exhibited survival rates similar to those of control GFP OE mice (Fig S1C). To examine whether YAP6SA impacts cardiac function in mice, we performed echocardiography to measure the ejection fraction (EF) and fractional shortening (FS) levels of GFP and YAP6SA OE hearts 4 or 8 wk after AAV9 delivery. YAP6SA OE mice exhibited similar FS levels (~40%) and EF levels (~75%) compared with AAV9-GFP control mice, and presented similar echocardiograms (Fig S1D and E). Together, these data suggest that YAP6SA overexpression is well tolerated in vivo and does not impair heart function in mammals.

### YAP6SA reprograms transcriptional networks in cardiomyocytes

To elucidate the mechanisms underlying YAP6SA function, we first examined changes in gene expression profiles in YAP6SA OE hearts, comparing them with both GFP controls and YAP5SA OE hearts. We isolated CM nuclei 3 d after AAV9 infection and extracted RNA for transcriptional profiling (Fig 2A). Among all groups, we identified five distinct clusters of differentially expressing genes (DEGs), where YAP6SA OE CMs had fewer DEGs compared with YAP5SA OE CMs, with 209 up-regulated genes and 96 down-regulated genes (FDR < 0.05) (Fig 2B and C). These data suggest that YAP6SA has decreased transcriptional activity compared with YAP5SA. In addition, gene ontology (GO) analysis revealed that YAP5SA-specific up-regulated genes are involved in mitotic cell cycle progression, and the down-regulated genes are related to oxidative metabolism (Fig 2D), whereas YAP6SA-specific up-regulated genes participate in TCA cycle and CM differentiation. Notably, YAP5SA and YAP6SA OE CMs shared DEGs that are relevant to cytoskeleton organization, and the GO analysis for YAP6SA versus GFP OE CMs confirmed the increased expression of actin and microtubule-related genes, such as *Acta1*, *Actg1*, *Myl9*, and *Rhoa* (Loumaye et al, 2022) (Fig 2F). In addition, we found that up-regulated genes, including *Myh8* and *Srf* (Fig 2E), mediate muscle cell differentiation and development (Aksel et al, 2015; Kwon et al, 2021) and others, such as *Ndufa1*, *Pdk4*, are TCA metabolic pathway genes (Yatsuka et al, 2020; Song et al, 2021) (Fig 2F and G). Moreover, down-regulated genes, such as *Csf1*, *Arel1*, are involved in the immune response (Chitu & Stanley, 2006; Lear et al, 2019). The predicted upstream regulators of DEGs in YAP6SA-overexpressing CMs included YAP itself, corroborating the RNA-seq findings and confirming YAP's canonical role as a transcriptional regulator (Fig 2E). Overall, these data suggest a TEAD-independent role of YAP6SA in mediating CM cellular activities in transcriptional or nontranscriptional manners.

### Metabolomics profiling of YAP6SA-overexpressing hearts

To determine whether YAP6SA alters the cardiac metabolic environment, we performed targeted metabolomics to evaluate glycolysis and TCA cycle–related metabolite levels in GFP control and YAP6SA OE hearts 5 d after AAV9 delivery into the P6 mice (Fig 3A). Compared with controls, YAP6SA OE hearts exhibited distinct metabolic profiles (Fig 3B). Given the involvement of these metabolites in the pentose phosphate pathway (PPP) (Fig 3C), we conducted a more detailed analysis and consistently observed elevated levels of glucose/fructose, G6P/F6P, ribose/ribulose/xylulose-5P, and ribulose-5P in YAP6SA OE hearts (Fig 3D).

Altogether, our data provide evidence that YAP6SA induces the activation of PPP in postnatal hearts. The PPP, also known as the phosphogluconate pathway or hexose monophosphate shunt, is a metabolic pathway that operates in parallel with glycolysis and produces NADPH and ribose-5-phosphate (TeSlaa et al, 2023). Notably, ribose-5-phosphate is a precursor for nucleotide synthesis. Hence, our data suggest that YAP6SA initiates a metabolic change that may associate with increased activity of DNA replication, which may indicate CM cell cycle reentry.

### YAP6SA promotes cardiomyocyte nucleation

To investigate whether YAP6SA stimulates CM division, we collected postnatal mouse hearts after 3 d of AAV9 infection and performed immunofluorescence (IF) to examine the CM cell cycle activity using the cell cycle markers CDK2, pHH3, and cyclin A2. CDK2 and cyclin A2 are highly expressed during the S phase of cell cycle (Israels & Israels, 2000). During the mitotic (M) phase, phosphorylated histone H3-Ser10 (pHH3) emerges from the onset of chromosome condensation (prophase) through metaphase and declines at anaphase (Sawicka & Seiser, 2012). IF results revealed that compared with AAV9-GFP control hearts, YAP6SA OE hearts exhibit increased CM proliferation activity, marked by ~10% CDK2-positive CMs (Fig 4A), 4.0% cyclin A2–positive CMs (Fig 4B), and 0.6% pHH3-positive CMs per area (Figs 4C and S2A). In addition, YAP6SA OE hearts had similar numbers of dividing CMs compared with YAP5SA OE hearts. We further traced CM division in control and YAP6SA OE postnatal hearts via six consecutive EdU injections after 2 d of AAV9 delivery and isolated CMs for analysis after a 48-h chasing period (Fig 4D). Consistently, YAP6SA OE hearts exhibited an increased number of EdU-positive CMs compared with the control (Fig 4E and F), revealing that YAP6SA stimulates CM cell cycle reentry. Also, the pH2AX staining revealed an enhanced DNA double-strand break response during DNA replication in YAP6SA-overexpressing CMs (Fig S2B).

Moreover, we quantified mono- and binuclear CMs in GFP control or YAP6SA OE neonatal hearts 7 d after AAV9 infection at P1 age and observed that YAP6SA OE hearts have an increased percentage of binuclear CMs (Fig 4G), confirming that YAP6SA promotes CM karyokinesis. To evaluate the DNA content of proliferated CMs, we performed flow cytometry for EdU-positive CMs in GFP OE and YAP6SA OE postnatal hearts (Fig 4H), and found that compared with controls, the EdU-positive CMs in YAP6SA OE mice have an increased percentage of 2N nuclei and a decreased percentage of 8N nuclei (Fig 4I). These data demonstrate that YAP6SA stimulates CMs to reenter the cell cycle and drives CMs to nucleate.

### YAP6SA interacts with multiple protein factors in cardiomyocytes

To identify YAP6SA interactors in CMs, we used a Flag antibody to pull down YAP6SA and its interacting proteins in mouse postnatal heart extracts 3 d after AAV9 delivery for mass spectrometry analysis. YAP5SA and YAP6SA share many protein interactors while having their unique binding factors (Fig 5A). YAP6SA pulled down diverse protein factors (Fig 5B), including the classical Hippo pathway components WWC1/KIBRA, AMOTL2, and NF2, which

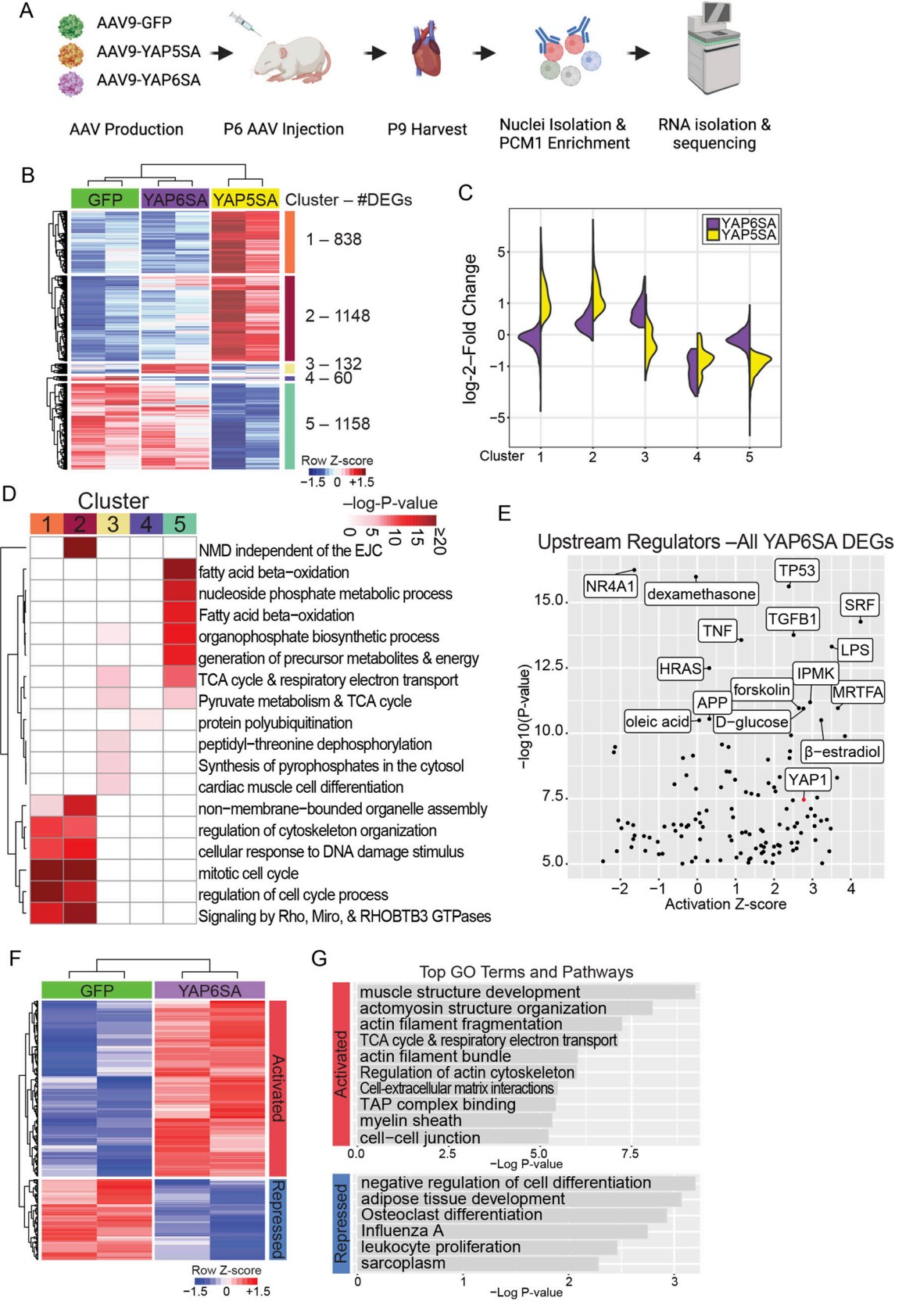

regulate LATS1/2 and YAP/TAZ activities (Zhao et al, 2011; Matsuda et al, 2016; Park et al, 2016). YAP6SA also interacted with components of the cell cycle progression and CM development pathways, including RBBP8, PSMD14, PPP1CA, and the transcription cofactor FHL2 (Yu & Chen, 2004; Tran et al, 2016; Ma et al, 2021; Li et al, 2022). Interestingly, YAP6SA complexes with the RNA transcription and translation factors MRPS33, HNRNPH1, RBMX, EEF1G, PRPS1, and PRPS1L3 (Christian & Spremulli, 2012; Palacios et al, 2017; Cho et al, 2018; Jing et al, 2019; Takahashi et al, 2020), suggesting an alternative function of YAP6SA in CMs. We also observed that COX7A2, NDUFA8, and PARK7, which participate in oxidative metabolism in mitochondria (Taira et al, 2004; Deng et al, 2018; Yatsuka et al, 2020), interact with YAP6SA. Moreover, the cytoskeleton and cell junction organizers MPDZ, MPP5, MARCKS, CTNNA1/αE-catenin, and ZYX (Hulsken et al, 1994; Kantardzhieva et al, 2006; Hirata et al, 2008; Assemat et al, 2013; Yu et al, 2015) bind to YAP6SA (Fig 5B). Given our mass-spec and RNA-seq data, we determined that YAP6SA has multiple TEAD-independent roles in regulating various CM activities, and given the increased ratio of YAP6SA in the cytoplasm, we hypothesize that YAP6SA has a central role in modulating the CM cytoskeleton structure to provide a permissive environment for CM cell cycle progression.

### YAP6SA promotes cardiomyocyte nucleation by activating Rho GTPases

Our mass spectrometry results demonstrated that the MPDZ protein is a significant interactor of YAP6SA, with a strong binding affinity. Also known as MUPP1, MPDZ contains 13 PDZ domains and serves as a scaffolding protein to organize higher order protein complexes and maintain cell polarity (Ullmer et al, 1998; Assemat et al, 2013). The YAP/MPDZ interaction has been identified in numerous studies (Yu & Guan, 2013; Park et al, 2016), and MPDZ has been shown to coordinate with Rho guanine nucleotide exchange factors (GEFs) to activate Rho GTPases during endothelial cell migration and in the vicinity of synapses (Estevez et al, 2008; Ernkvist et al, 2009). Rho family proteins mediate almost all fundamental cellular processes in eukaryotes. Importantly, they regulate cytoskeleton reorganization during cytokinesis in certain cell types (Piekny et al, 2005; Mosaddeghzadeh & Ahmadian, 2021). Given the limited understanding of how the YAP/MPDZ complex regulates cardiomyocyte proliferation, we investigated whether YAP6SA promotes cardiomyocyte cell cycle reentry through its interaction with MPDZ, potentially enhancing Rho protein activity (Fig 5D). We performed co-IP to validate the YAP6SA-MPDZ interaction in postnatal CMs (Fig 5C), and evaluated the expression levels of Rho family genes among GFP, YAP5SA, and YAP6SA OE groups based on our bulk RNA-seq result. Surprisingly, the Rho genes, notably *Rhobtb1* and *Rhoa*, were up-regulated in YAP6SA OE groups (Fig 5E).

To determine whether inhibiting MPDZ expression in YAP6SA OE hearts would reduce Rho activity, we infected P6 YAP6SA hearts with AAV9 carrying MPDZ-targeted GFP-shRNA and collected hearts a week later. Western blotting revealed that MDPZ inhibition reduced Rho protein levels (Fig 5F and G). We also examined CM cell cycle activity 3 d after AAV9 delivery (Fig 5H). IF images revealed that knocking down MPDZ reduces CDK2-positive CMs in YAP6SA OE hearts (Fig 5I). Using an in vitro assay, we further observed that inhibiting MPDZ in iPS-CMs decreases Aurora B (broadly detected throughout the M phase and cytokinesis [Sawicka & Seiser, 2012]) and Rho A levels in YAP6SA OE iPS-CMs (Fig S3A–C). The Flag and MPDZ localization staining also suggested the interaction between YAP6SA and MPDZ in iPS-CMs (Fig S3D). To determine whether suppressing Rho GTPases activity impairs YAP6SA function, we administered the ROCK inhibitor Y-27632 in YAP6SA OE postnatal hearts and quantified CDK2- and pHH3-positive CMs 2 d later (Fig 5J). Interestingly, we found that CDK2- and pHH3-positive CMs were significantly fewer in YAP6SA OE hearts treated with Y-27632 compared with DMSO controls (Fig 5K and L). Using neonatal rat ventricular myocytes (NRVMs) as an in vitro assay, we confirmed that inhibiting Rho GTPases activity by the ROCK inhibitor decreases the number of pHH3-positive NRVMs (Fig S3E). These data reveal that YAP6SA enhances CM cell cycle activity by interacting with MPDZ and activating Rho GTPases.

## Discussion

Mammalian hearts maintain their regenerative capacity for a short time after birth (Porrello et al, 2011). A long-standing question is how cardiac renewal is regulated during heart maturation. Our lab discovered that the HSP inhibits CM renewal through inhibiting the downstream effector YAP (Leach et al, 2017). We also identified YAP5SA, a modified YAP variant that bypasses Hippo inhibitory regulation, leading to CM hyperplasia, thickened ventricular walls, and, ultimately, mortality (Monroe et al, 2019; Li et al, 2024; Morikawa et al, 2025). Understanding how YAP5SA promotes cardiomyocyte proliferation is crucial for optimizing this active YAP variant as a potential therapy for heart failure.

We identified YAP6SA, an active YAP variant with disrupted YAP/TEAD interaction, that stimulates cardiomyocyte proliferation through TEAD-independent mechanisms. Compared with YAP5SA OE mice, YAP6SA extends survival while preserving cardiac structure and function. Notably, YAP6SA promotes cardiomyocyte cell cycle progression, as evidenced by a significant increase in proliferating cardiomyocytes in YAP6SA OE hearts compared with controls.

To uncover the mechanism(s) through which YAP6SA promotes CM proliferation, we combined RNA-seq and mass spectrometry analysis to identify DEGs and YAP6SA interactors in YAP6SA OE CMs. Unlike YAP5SA OE hearts, cell cycle–related gene expression is not

**Figure 2. YAP6SA drives widespread transcriptional changes in cardiomyocytes.**
**(A)** Bulk RNA-seq pipeline. **(B)** Heatmap of DEGs in CMs of indicated groups. **(C)** Split violin plot showing average fold change for each gene cluster in YAP5SA versus YAP6SA OE CMs. **(D)** GO analysis of DEGs in YAP5SA and YAP6SA OE CMs. **(E)** Predicted upstream regulators of DEGs in YAP6SA OE CMs. **(F, G)** Heatmap of DEGs and (G) GO analysis between GFP and YAP6SA OE CMs.
Source data are available for this figure.

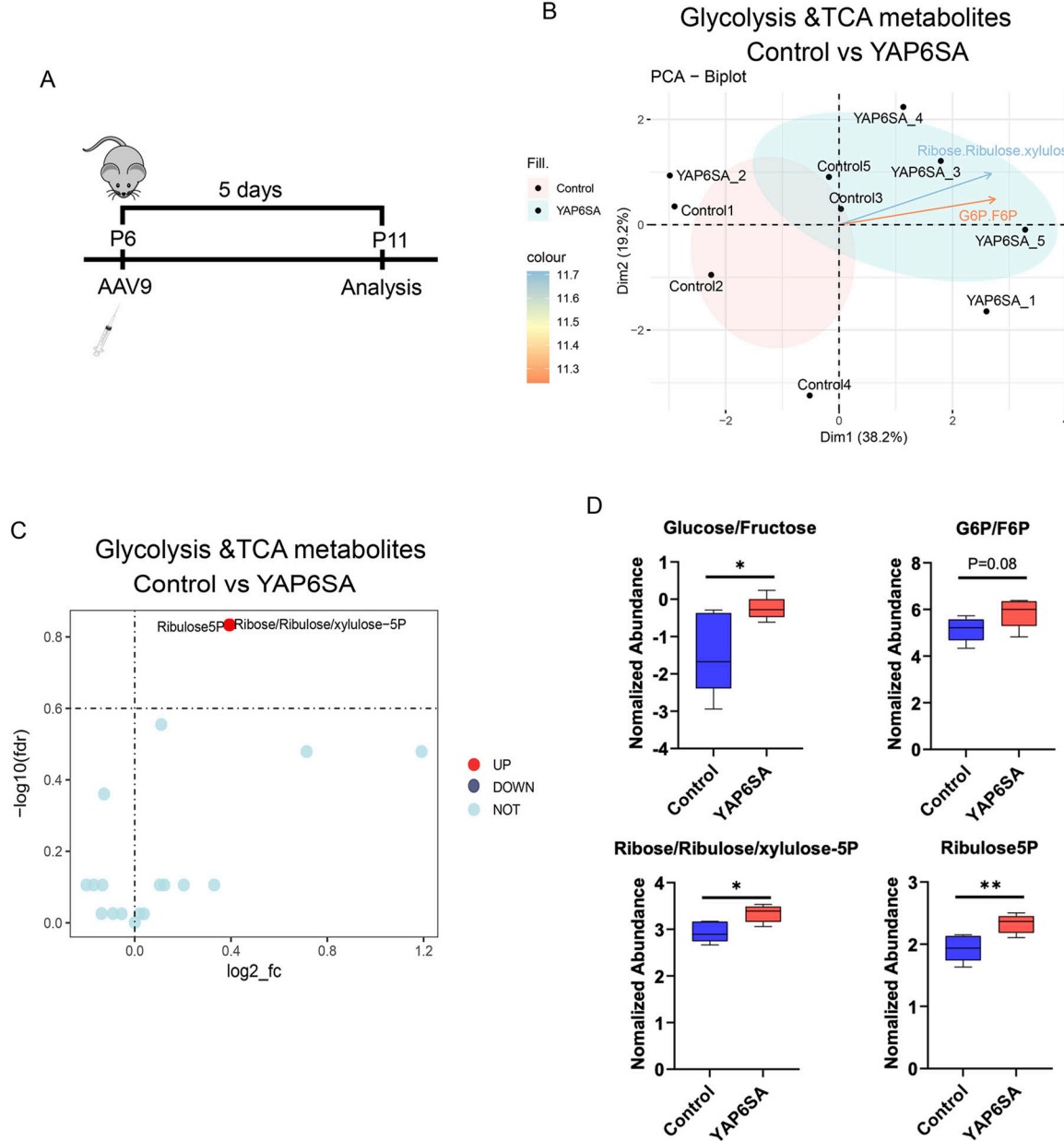

**Figure 3. Metabolomics profiling of YAP6SA-overexpressing hearts.**
**(A)** Time scheme for metabolomics analysis in P6 mouse hearts. **(B)** PCA plot showing the metabolomics difference of GFP control and YAP6SA OE hearts.
**(C)** Identification of differential metabolites in control and YAP6SA OE hearts. **(D)** Quantification of differential metabolites in control and YAP6SA OE hearts.
Source data are available for this figure.

up-regulated in YAP6SA OE CMs compared with controls. Interestingly, multiple cytoskeleton remodeling genes are up-regulated in YAP6SA OE CMs. In addition, we observed that YAP6SA interacts with skeletal proteins, such as MPDZ. This suggests that YAP6SA promotes CM division by overcoming mitotic barriers via CM cytoskeletal remodeling.

It is well established that Rho GTPases are essential for cell proliferation–associated cytoskeletal reorganization (Piekny et al, 2005). In this study, we observed that Rho family gene expression is elevated in dividing cardiomyocytes. Furthermore, we found that

silencing MPDZ in YAP6SA-overexpressing hearts led to reduced Rho protein expression. In addition, treatment with a ROCK inhibitor, which blocks the RhoA downstream effector ROCK, significantly decreases the number of CDK2- and pHH3-positive cardiomyocytes in YAP6SA OE hearts. These findings collectively reveal a critical role of Rho GTPase signaling in promoting cardiomyocyte cell cycle progression. The rationale by which the YAP6SA/MPDZ complex activates Rho GTPases to regulate CM cytoskeleton structure and drive CM mitosis remains unclear. Although previous studies have established the role of MPDZ in

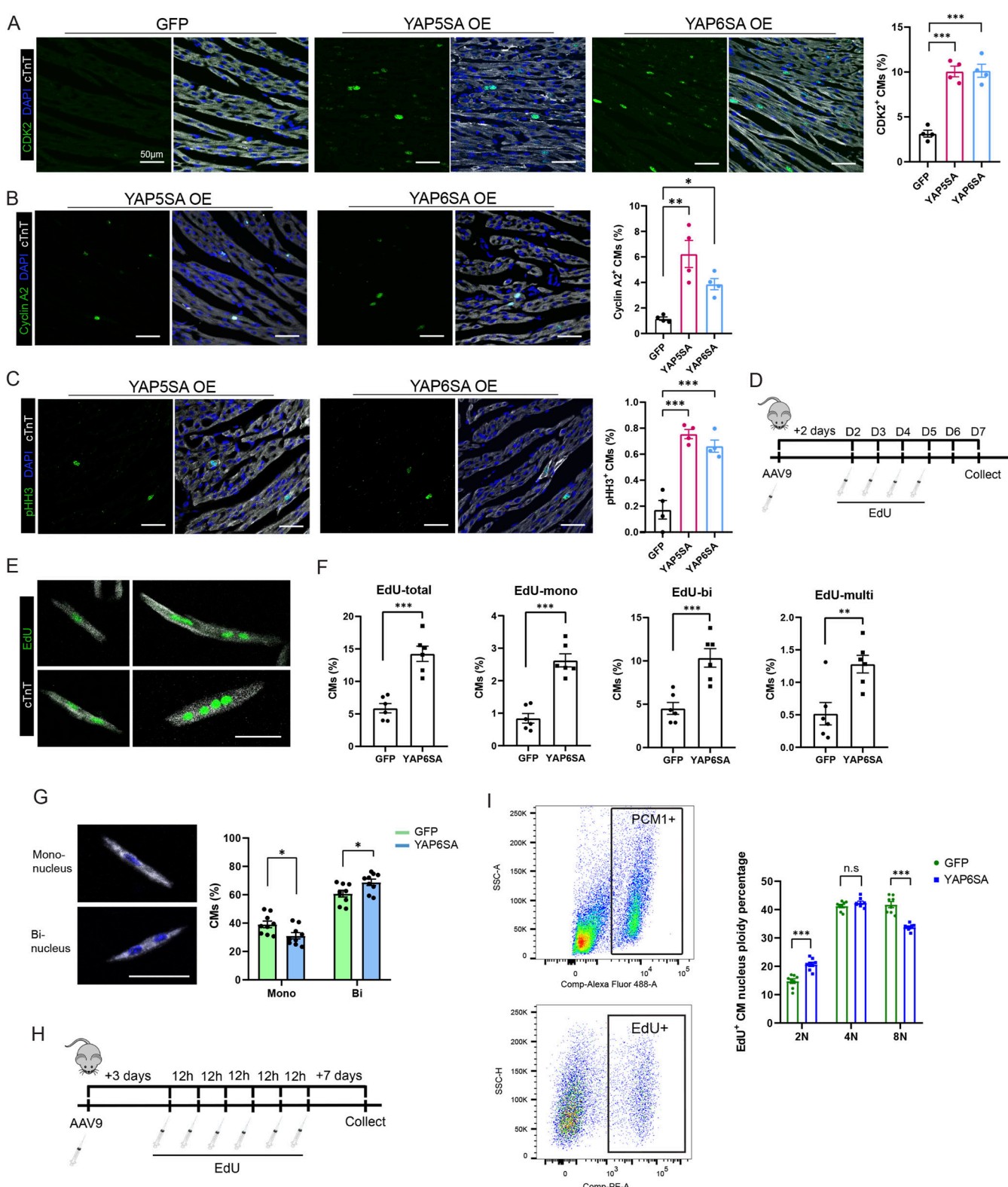

**Figure 4. YAP6SA promotes cardiomyocyte nucleation.**
**(A)** Immunofluorescence (IF) for CDK2-positive CM quantification (n = 4 each) after 3 d of AAV9 infection in P6 murine hearts. **(B)** IF for cyclin A2–positive CM quantification (n = 4 each) after 3 d of AAV9 infection in P6 murine hearts. **(C)** IF for pHH3-positive CM quantification (n = 4 each) after 3 d of AAV9 infection in P6 murine hearts. *$P < 0.05$, **$P < 0.01$, ***$P < 0.001$. **(D)** Time scheme of the EdU administration experiment in P6 mice. **(E)** IF images showing different types of EdU-positive CMs with mononucleus, binucleus, or multiple nuclei. **(F)** Quantification results for EdU-positive CMs (n = 500, N = 6). **(G)** Images showing mononucleus and binuclear CMs and quantification results for mononucleus and binuclear CMs in GFP and YAP6SA OE hearts after 7 d of AAV9 infection in P1 neonatal hearts (n = 500, N = 9). **(H)** Time scheme

facilitating the conversion of Rho-GDP to Rho-GTP (Estevez et al, 2008), our findings reveal that MPDZ also regulates the overall levels of Rho family proteins in cardiomyocytes, a relationship that has not been previously reported. One possible explanation for the observed reduction in Rho protein levels is the instability of Rho-GDP, as Rho guanine nucleotide dissociation inhibitors (RhoGDIs) are known to bind Rho-GDP and protect it from degradation (Boulter et al, 2010). In addition, multiple Rho proteins could be activated by YAP6SA/MPDZ, as the Rho family participates in various cell division events. Notably, recent studies have identified sarcomere disassembly as a prerequisite for cardiomyocyte cell cycle reentry (Liu et al, 2025; Morikawa et al, 2025); however, the mechanisms governing this process remain unclear, as does the specific involvement of cytoskeletal components directly linked to Rho GTPase activity.

Our study is the first to shed light on the TEAD-independent function of active YAP in connecting CM cytoskeleton remodeling to CM cell cycle progression. Mature CM proliferation is a complex and not yet fully understood process. We are actively investigating the novel ways YAP6SA regulates CM division, and are working to decipher the complex interplay between Rho GTPases and cytoskeleton reorganization during CM mitosis. These findings hold significant promise for improving cardiovascular outcomes in humans.

# Materials and Methods

### Animals

Mice were housed and maintained in accordance with recommendations set in the Guide for the Care and Use of Laboratory Animals of the National Institutes of Health. All animal protocols were approved by the Baylor College of Medicine Institutional Animal Care and Use Committee (IACUC). Male and female mice were used for all experiments. Mice were maintained on a FVB or ICR background. All control animals were littermates or age-matched if littermates were unavailable.

### AAV9 virus

The constructs containing rtTA, GFP, YAP5SA, YAP6SA, and shRNA targeting MPDZ gene sequences (Table S1) were cloned into a pENN.AAV.cTNT vector, which is transcribed under the cTnT promoter. All vectors were packaged into the muscle-trophic serotype AAV9 by the Intellectual and Developmental Disabilities Research Center Neuroconnectivity Core at the Baylor College of Medicine. After titering, viruses were aliquoted and immediately frozen and placed at −80°C for long-term storage. For the subcutaneous injection into neonatal and postnatal mice, each aliquot was diluted in saline to create a 50 $\mu l$ injection solution, delivering a total of $1 \times 10^{11}$ viral genomes to each mouse.

### Co-immunoprecipitation

Mouse whole hearts were homogenized and lysed using RIPA lysis buffer (10 mM Tris-Cl at pH 8.0, 140 mM NaCl, 1 mM EDTA, 1% Triton X-100, 0.1% sodium deoxycholate, 0.1% SDS, 1× protease inhibitor cocktail, and 1× phosphatase inhibitor [Roche]). Lysates were centrifuged at 13,800$g$ for 20 min, and supernatants were collected for immunoprecipitation. YAP5SA and YAP6SA and their interacting proteins were purified using Anti-FLAG M2 Magnetic Beads (Sigma-Aldrich) for 4 h of rotated incubation at 4°C. The beads were washed three times, 10 min each, using RIPA lysis buffer and boiled with elution buffer (4 × loading: RIPA = 1:3) for 10 min. The antibodies used for immunoblotting in this context were rabbit anti-DYKDDDDK Tag (D6W5B) (1:2,000), Cat#14973; Cell Signaling Technology; rabbit anti-YAP (1:2,000), Cat#NB110-583538; Novus Biologicals; rabbit anti-TEADs (D3F7L) (1:1,000), Cat#13295; Cell Signaling Technology; rabbit anti-MUPP1/MPDZ (1:1,000), Cat#42-2700; Invitrogen.

### Western blotting

Western blotting was performed using standard methods with lysates prepared by homogenizing hearts with a homogenizer in RIPA buffer. The lysates, after 5 min of boiling in a reducing Tris-based SDS sample buffer, were loaded into acrylamide gels and run at 120 V for a sufficient time to achieve separation. Proteins were then transferred to PVDF membranes and imaged using the Amersham Imager 680 system (GE Healthcare). Primary antibodies were as follows: rabbit anti-DYKDDDDK Tag (D6W5B) (1:2,000), Cat#14973; Cell Signaling Technology; rabbit anti-GAPDH (1:3,000); Abcam; mouse anti-$\beta$-actin (C4) (1:3,000), Cat#sc-47778; Santa Cruz Biotechnology; rabbit anti-MUPP1/MPDZ (1:1,000), Cat#42-2700; Invitrogen; rabbit recombinant anti-Rho A + B + C antibody (EPR18299) (1:1,000), Cat#ab188103; Abcam. HRP-conjugated secondary antibodies were goat anti-rabbit IgG (H+L) and goat anti-mouse IgG (H+L) (1:5,000), Cat#111-035-003; Jackson Immuno-Research. Quantitation was performed using the gel analysis feature in Fiji (ImageJ) (National Institutes of Health, Bethesda, MD, USA).

### Ultrasound echocardiography

M- and B-mode parasternal echocardiography of the left ventricle was performed according to established protocols at the Baylor College of Medicine Mouse Phenotyping Core (Respress & Wehrens, 2010) using the MS550S transducer operating at 40 MHz on a VisualSonics Vevo 2100 system and analyzed using Vevolab 5.7 software (Fujifilm VisualSonics).

### Histology and immunofluorescence

Freshly dissected hearts were imaged for GFP fluorescence using a Zeiss LSM 780 confocal microscope. For fixation, hearts were

---

of EdU administration experiment for ploidy analysis in P6 murine hearts. **(I)** Flow cytometry results showing the PCM-1 and EdU selection on replicated CM nuclei and the percentage of ploidy in GFP and YAP6SA OE hearts (n = 8).

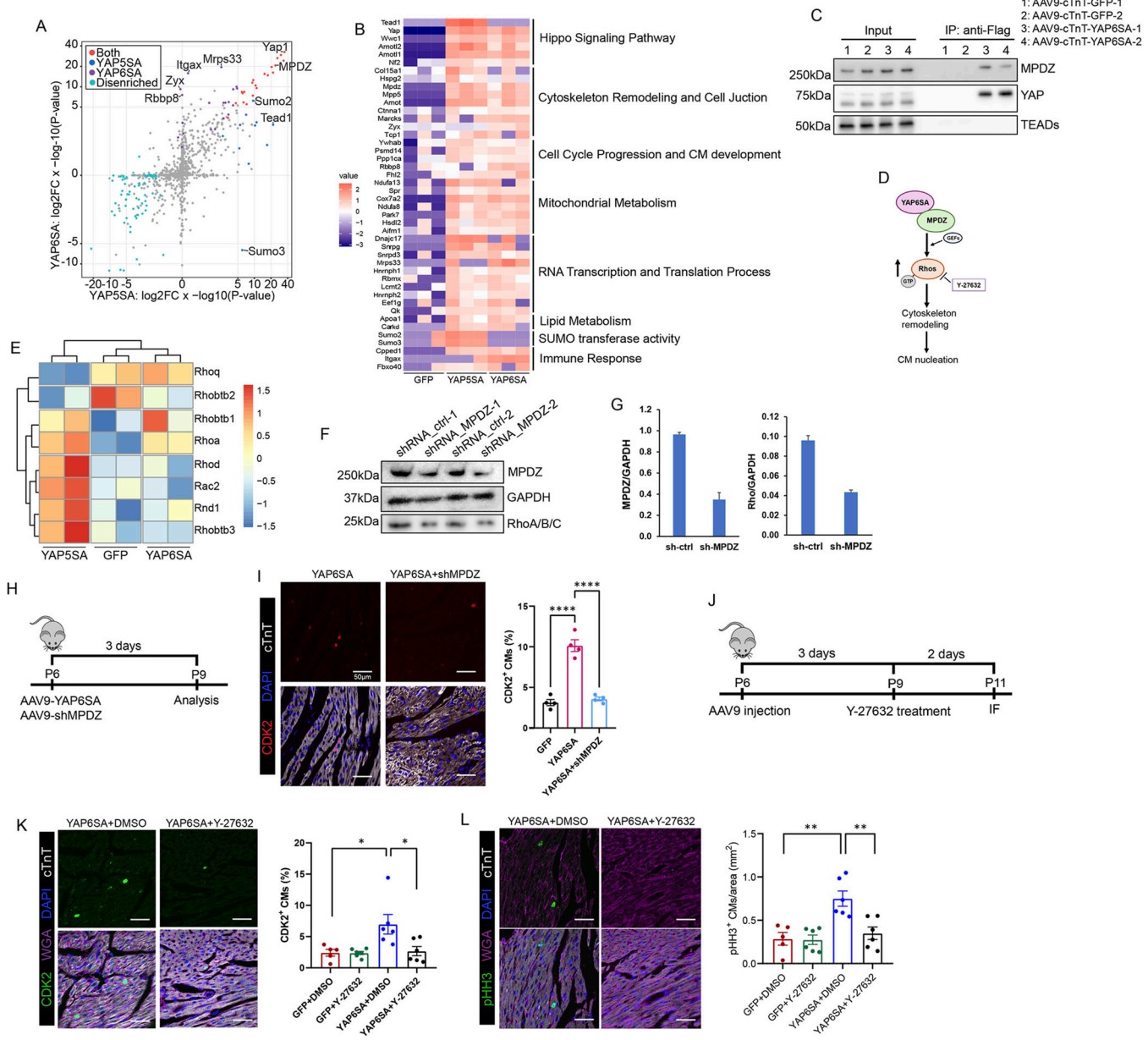

**Figure 5. YAP6SA interacts with a diverse range of protein factors in cardiomyocytes.**
**(A)** Plot of YAP5SA and YAP6SA interactome data in CMs after 3 d of AAV9 infection in P6 mice. **(B)** Classification of YAP5SA and YAP6SA interactors in CMs. **(C)** IP result showing YAP6SA interaction with MPDZ in CMs after 3 d of AAV9 infection in P6 mice. **(D)** Graphic showing the hypothesized mechanism of YAP6SA function. **(E)** Bulk RNA-seq data demonstrating increased Rho mRNA levels in YAP6SA OE hearts. **(F, G)** Western blot showing decreased Rho protein levels in CMs expressing MPDZ-targeted shRNA and the quantification result after 7 d of AAV9 infection in P6 murine hearts. The shRNA is transcribed with GFP. **(H)** Time scheme showing AAV9-YAP6SA and AAV9-shMPDZ delivery to P6 mouse hearts and analysis 3 d post-AAV9. **(I)** CDK2 staining and quantification of YAP6SA and YAP6SA with shMPDZ hearts. **(J)** Time scheme for ROCK inhibitor treatment in P6 postnatal hearts. **(K)** IF staining and quantification results comparing the number of CDK2-positive CMs among groups (n = 6 each). **(L)** IF staining and quantification results comparing the number of pHH3-positive CMs among groups (n = 6 each).
Source data are available for this figure.

retrogradely perfused with 20 mM KCl-PBS before being perfused with 10% neutral buffered formalin, followed by embedding in paraffin. Transverse sections (seven microns) were cut and mounted onto charged polylysine slides. A portion was stained with Masson's trichrome stain or H&E staining. Immunohisto-chemistry was performed by first deparaffinizing and rehydrating

sections, followed by antigen retrieval and permeabilization in 0.5% Triton X in PBS. Sections were blocked (10% donkey serum in PBS, 0.1% Triton X) and then incubated with primary antibody overnight at 4°C, and secondary antibodies for 1 h at RT before imaging (Figs 4A and 5I and K, rabbit anti-CDK2, Cat#32147; Abcam; anti-rabbit Alexa 488, Cat#A-21206; Thermo Fisher Scientific; anti-

rabbit Alexa 555, Cat#A-31572; Thermo Fisher Scientific; mouse anti-cTnT-Alexa 647 conjugate, Cat#565744; BD Pharmingen) (Fig 4B, rabbit anti-cyclin A2, Cat#32386; Abcam; anti-rabbit Alexa 488, Cat#A-21206; Thermo Fisher Scientific) (Figs 4C and 5L, rat anti-pHH3, Cat#ab10543; Abcam; anti-rat Alexa 488, Cat#A-21208; Thermo Fisher Scientific) (Fig S2B, rabbit anti-pH2AX, Cat#9718S; Cell Signaling Technology; anti-rabbit Alexa 555, Cat#A-31572; Thermo Fisher Scientific). Rhodamine-conjugated WGA was from Vector Labs, Cat#RL-1022. Nuclei were stained with DAPI (Cat#62248; Thermo Fisher Scientific). All imaging was performed with a Zeiss LSM 780 confocal microscope and 20× (Air (dry), ~0.8) or 40x objectives (Oil immersion, ~1.4) in the Optical Imaging and Vital Microscopy Core at Baylor College of Medicine.

For frozen sections, hearts were dehydrated with 15% and 30% sucrose/PBS solution and then placed into Tissue-Tek optical cutting temperature (OCT) compound (V.W.R. Cat#25608-930) before freezing over dry ice. Sections (10 microns) were then cut and mounted on glass slides. For immunofluorescence staining, sections were fixed and permeabilized and then incubated with primary and secondary antibodies before imaging (Figs 1E and S1A, rabbit DYKDDDDK Tag Antibody, Cat#14793; Cell Signaling Technology; mouse anti-cTnT-Alexa 647 conjugate, Cat#565744; BD Pharmingen) (Fig S2A, rat anti-pHH3, Cat#ab10543; Abcam; anti-rat Alexa 488, Cat#A-21208; Thermo Fisher Scientific). Rhodamine-conjugated WGA was from Vector Labs, Cat#RL-1022. Nuclei were stained with DAPI.

For in vitro staining, cells were fixed in 4% PFA with PBS for 20 min at RT, then permeabilized and blocked in PBS with 0.2% Triton X-100, 0.1% Tween-20, and 3% donkey serum for 30 min, and then incubated with primary and secondary antibodies before imaging. (Fig S1B, rabbit DYKDDDDK Tag Antibody, Cat#14793; Cell Signaling Technology) (Fig S3A–D, rabbit anti-Aurora B, Cat#ab2254; Abcam; anti-rabbit Alexa 488, Cat#A-21206; Thermo Fisher Scientific; rabbit anti-Rho A antibody, Cat#ab86297; Abcam; rabbit MUPP1/MPDZ polyclonal antibody, Cat#42-2700; Invitrogen; rabbit DYKDDDDK Tag Antibody, Cat#14793; Cell Signaling Technology) (Fig S3E, rabbit recombinant anti-Rho A + B + C antibody [EPR18299] (1:1,000), Cat#ab188103; Abcam; rat anti-pHH3, Cat#ab10543; Abcam).

## EdU labeling analysis

EdU was dissolved in PBS. Mice were injected with EdU intraperitoneally for several days (0.5 mg/g), and their hearts were collected after a chase period. The hearts were fixed in 4% PFA with PBS for 1 h at RT, then minced into small pieces, and incubated with Collagenase B and D in HBSS overnight. Isolated CMs were permeabilized in PBS with 0.2% Triton X-100 for 20 min and then washed by PBS with 0.1% Tween-20 (PBST) followed by EdU solution incubation for 30 min at RT. Then, CMs were incubated with anti-cTnT-647 antibody in 0.1% PBST at 4°C overnight and later stained with DAPI. Quantification of EdU-positive CMs was then performed.

## Nuclear isolation for sequencing

Nuclear isolation was performed as previously described with the following specifications (Monroe et al, 2019): Briefly, fresh cardiac tissue was harvested on ice and was immediately cut into tiny pieces before performing Dounce homogenization in NP-40 lysis buffer (10 mM Tris–HCl, pH 7.4, 10 mM NaCl, 3 mM MgCl$_2$, 0.1% NP-40, 1 mM DTT, and RNase inhibitors). Homogenized solution was filtered, and nuclei were isolated via density gradient centrifugation with OptiPrep density gradient medium after mixing homogenate 1:1 with a working solution (5 volumes of OptiPrep [Cat#D1556; Sigma-Aldrich] with 1 volume of diluent [20 mM MgCl$_2$, 60 mM Tris-Cl, pH 7.4, 50 mM NaCl, 6% BSA, 6 mM DTT, and RNase inhibitors]). After 12-min 10,000 G centrifugation, all nuclei isolated from a 30–40% interface were precleared with Protein G Dynabeads (Cat#10003D; Thermo Fisher Scientific). Next, nuclei were immunoprecipitated with an anti-PCM-1 (Cat#HPA023370; Sigma-Aldrich) antibody and Protein G Dynabeads (washing two times with wash buffer [10 mM Tris–HCl, pH 7.4, 10 mM NaCl, 3 mM MgCl$_2$, 1% BSA, 0.1% Tween-20, 1 mM DTT, and RNase inhibitors]) to enrich for CM nuclei as described previously (Purdy et al, 2023).

## RNA sequencing

RNA from bead-bound PCM-1(+) nuclei was collected using the RNeasy Plus Micro kit (QIAGEN). Poly-A–enriched nuclear RNA sequencing (RNA-seq) libraries were generated by Novogene America and sequenced on the Illumina NovaSeq 6000. Reads were mapped to the mouse genome (mm10, Ensembl GRCm38.94) using STAR (Dobin et al, 2013). Differential expression analysis was performed using edgeR (Robinson et al, 2010). Lowly expressed genes were removed within each condition using median log$_2$-transformed counts per gene per million mapped reads of 1, and a union set of genes was generated from each condition. Differential expression analysis was performed using a general linear model framework with an additional covariate to mitigate an apparent batch effect. Gene ontology analysis was performed using Metascape (Tripathi et al, 2015) and Ingenuity Pathway Analysis (QIAGEN; Kramer et al, 2014).

## Flow cytometry analysis of DNA content

EdU was administered, and CM nuclei were isolated as previously described. Hearts were dissected and minced on ice into NP-40 lysis buffer (10 mM Tris–HCl, pH 7.4, 10 mM NaCl, 3 mM MgCl$_2$, 0.1% NP-40, 1 mM DTT, and RNase inhibitors). PCM-1–bound nuclei were then conjugated to fluorescent secondary antibodies, and EdU was detected using Click-iT EdU Flow Cytometry Assay Kit according to the manufacturer's instructions (Cat#C10340; Thermo Fisher Scientific). DNA content was quantified by DAPI fluorescence. Detection was performed using a BD FACSAria II (BD Biosciences), and data were analyzed in FlowJo software (Tree Star). Gating was based on isotype controls and fluorophore-negative controls. Flow cytometry was performed at the Texas Heart Institute Flow Cytometry Core Facility.

## Targeted metabolomics analysis

To identify differentially abundant metabolites in YAP6SA versus GFP control hearts, AAV9-cTnTGFP and AAV-cTNT-YAP6SA were injected into P6 WT postnatal mice, and hearts were collected 5 d after injection. Dissected hearts were then perfused, weighed,

snap-frozen in liquid nitrogen, stored in –80°C, and sent to the Mouse Metabolomics Core at Baylor College of Medicine for analysis. The targeted metabolomics study was performed focusing on glycolysis, TCA cycle, carnitines, and fatty acid metabolites (Kami Reddy et al, 2024). Briefly, the acquired data were analyzed, and review and integration of each peak, along with each of isotopomer peaks for metabolic flux, were done using Agilent Mass Hunter Quantitative Analysis software (Agilent Technologies). The data were $log_2$-transformed followed by internal standard normalization on a per-sample, per-method basis. For every metabolite in the normalized dataset, two-tailed $t$ tests were conducted to compare expression levels between AA BLCA and EA BLCA. Differential metabolites were identified by adjusting the $P$-values for multiple testing at an FDR threshold of < 0.25. The R statistical software system generated a hierarchical cluster of the differentially expressed metabolites (https://www.r-project.org/).

### iPS cell culture and differentiation

Cells were cultured on Geltrex-coated plates and fed with mTeSR Plus medium every other day. For cell differentiation, iPSCs were seeded as single cells onto Geltrex-coated plates and cultured in mTeSR Plus for 2 d until they reached ~80% confluency. Cells were then treated with CHIR99021 for 24 h, followed by C59 for 48 h, and subsequently maintained in RPMI/B27 medium. Nonmyocytes were eliminated using DMEM supplemented with 4 mM lactate. The purified cardiomyocytes were then matured in GFAM for 3 wk.

### Mass spectrometry

IP beads were resolved on NuPAGE 10% Bis-Tris Gel with MOPS running buffer (Life Technologies). The eluted proteins were visualized with Coomassie brilliant blue stain, excised into gel pieces, and in-gel–digested with trypsin. The heavy and light chain bands were pooled together, whereas the remaining MW region was combined to generate another pool. The LC-MS/MS analysis was carried out using nanoLC1000 system coupled with Orbitrap Fusion mass spectrometer (Thermo Fisher Scientific). The peptides were loaded on a two-column setup with precolumn (2 cm × 100 $\mu$m I.D.) and analytical column (20 cm × 75 µm I.D.) filled with ReproSil-Pur Basic-C18 (1.9 $\mu$m; Dr. Maisch GmbH). The peptide elution was done using a discontinuous gradient of 90% acetonitrile buffer (B) in 0.1% formic acid (2–30% B: 86 min, 30–60% B: 6 min, 60–90% B: 8 min, 90–50% B: 10 min). The MS instrument was operated in data-dependent mode with MS1 acquisition in Orbitrap (120,000 resolution, AGC 5 × $10^5$, 50-ms injection time) followed by MS2 in Ion Trap (Rapid Scan, HCD 30%, AGC 5 × $10^4$). The MS raw data were searched using Proteome Discoverer 2.1 software (Thermo Fisher Scientific) with Mascot algorithm against mouse NCBI RefSeq database updated 2020_0324. The precursor ion tolerance and product ion tolerance were set to 20 ppm and 0.5 Da, respectively. Maximum cleavage of 2 with trypsin enzyme, dynamic modification of oxidation on methionine, protein Nterm acetylation, and des-treak on cysteine were allowed. The peptides identified from Mascot result file were validated with 5% false discovery rate (FDR). The gene product inference and quantification were done with a label-free iBAQ approach using the "gpGrouper" algorithm (PMID:

30093420). For statistical assessment, missing value imputation was employed through sampling a normal distribution N ($\mu$-1.8 $\sigma$, 0.8 $\sigma$), where $\mu$, $\sigma$ are the mean and SD of the quantified values. For differential analysis, we used the moderated $t$ test and $log_2$ fold changes as implemented in the R package limma.

### In vivo drug treatment

For the in vivo treatment experiments, the ROCK inhibitor Y-27632 (Inc. 72304; StemCell Technologies) was diluted in mineral oil (Lot# MKCH0156; Sigma-Aldrich) at a dosage of 10 $\mu$g/g through subcutaneous injection into P9 mice. Mouse hearts were collected 2 d after drug treatment and were analyzed via pHH3 immunofluorescence followed by quantification.

### Quantification and statistical analysis

All statistical tests, error bars, $P$-values, and n numbers are reported in the corresponding Fig legends. Sample sizes were not predetermined but were chosen based on previous publications. Mice were excluded only if they had obvious anatomical or health abnormalities before experiments were performed. To address randomness, all available mice (mutant or control) were included in the study. Control mice are indicated in the figure legends. They were AAV9-GFP–infected mice; AAV9-rtTA–infected mice; or mice without AAV9 virus infection. Controls were littermates with or age-matched to experimental mice. No differences in variances were detected between any group in the reported experiments. One-way, two-tailed analysis of variance (ANOVA) tests, followed by post hoc tests, were performed in Origin Pro (OriginLab Corporation). Fisher's exact tests, chi-squared test, and log-rank tests were performed in Prism 5 (GraphPad). All graphs were generated in R or Microsoft Excel and presented using Prism. Cartoons were created in BioRender (https://www.biorender.com/; https://www.R-project.org/).

## Data Availability

Cell lines, plasmids, and other materials are available upon request. All raw and processed sequencing data are deposited at the National Center for Biotechnology Information's Gene Expression Omnibus (GEO): GSE315945. Source Data are provided with the article.

## Supplementary Information

## Acknowledgements

The Mouse Metabolism and Phenotyping Core at Baylor College of Medicine assisted with this project, which is supported in part by funding from the NIH (UM1HG006348, R01DK114356), and the use of an instrument purchased

with NIH funding (S10OD032380). BCM Mass Spectrometry Proteomics (RRID: SCR_027015) and Metabolomics Core also supported this work by the Dan L Duncan Comprehensive Cancer Center Award (P30 CA125123) and CPRIT Core Facility Awards (RP210227). We thank TT Tran of Baylor College of Medicine for assistance with animal procedures and L Zhang of Baylor College of Medicine for providing iPS-CMs. We are also thankful to Alon Azares, who directs the Texas Heart Institute Flow Cytometry Core Facility, for supporting this work.

## Author Contributions

B Xie: conceptualization, data curation, formal analysis, investigation, methodology, project administration, and writing—original draft.

J Steimle: data curation, formal analysis, funding acquisition, investigation, and methodology.

V Deshmukh: formal analysis, investigation, and methodology.

L Liu: formal analysis, funding acquisition, investigation, and methodology.

C-R Tsai: methodology.

TR Heallen: conceptualization and writing—review and editing.

W Paltzer: formal analysis, funding acquisition, and methodology.

Y Morikawa: methodology.

F Meng: funding acquisition and investigation.

J Wang: investigation.

JF Martin: conceptualization, formal analysis, supervision, funding acquisition, and writing—review and editing.

## Conflict of Interest Statement

JF Martin is a founder and owns shares in Yap Therapeutics, Inc. JF Martin is a coinventor on the following patents associated with this study: patent no. US20200206327A1 entitled "Hippo pathway deficiency reverses systolic heart failure post-infarction," patent no. 15/642200.PCT/US2014/069349 101191411 entitled "Hippo and dystrophin complex signaling in cardiomyocyte renewal," and patent no. 15/102593.PCT/US2014/069349 9732345 entitled "Hippo and dystrophin complex signaling in cardiomyocyte renewal."

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
