## [Reviewer comments · Life Science Alliance]

TEAD-independent Mechanisms of YAP Function in Cardiomyocytes Cell Cycle Re-entry

Bing Xie, Jeffrey Steimle, Vaibhav Deshmukh, Lin Liu, Chang-Ru Tsai, Todd Heallen, Wyatt Paltzer, Yuka Morikawa, Fansen Meng, Jun Wang, and James Martin

DOI: <https://doi.org/10.26508/lsa.202503496>

Corresponding author(s): James Martin, Baylor College of Medicine

Review Timeline:

Submission Date:	2025-08-27
Editorial Decision:	2025-11-06
Revision Received:	2025-11-20
Editorial Decision:	2026-01-06
Revision Received:	2026-01-12
Accepted:	2026-01-13

Scientific Editor: Sarita Hebbar

Transaction Report:

November 6, 2025

Re: Life Science Alliance manuscript #LSA-2025-03496

Dr. James F. Martin
Baylor College of Medicine
One Baylor Plaza
Houston, TX 77030

Dear Dr. Martin,

Thank you for submitting your manuscript entitled, "TEAD-independent Mechanisms of YAP Function in Cardiomyocytes" to Life Science Alliance. Your manuscript was evaluated by three expert reviewers whose comments are appended below. As you will read, all the reviewers have found your study interesting and of potential value. But they have also raised significant concerns that preclude publication at this stage, and have made suggestions to address these.

1. The reviewers have highlighted the lack of evidence supporting some claims in this work. For example, metabolic changes benefit cell cycle entry, YAP6SA having a 'TEAD-independent' role using non-transcriptional mechanisms, and the physiological role of YAP6SA-positivity in cardiomyocytes (versus over-expression). We concur with the reviewers that you must provide experimental evidence to back such claims. In the absence of this, you must tone down your conclusions.
2. All the reviewers suggest that you elaborate on the interpretation and discussion of the results in the context of (a) published findings and (b) known attributes of the proteins in question (for example in the context of normal expression, function of YAP, its sub-cellular localisation). We agree that this must be included in a revised submission.
3. The reviewers also point to missing or confusing details in the methods section and the lack of an accompanying rationale for described experiments, which we agree you must clarify or provide in a revised submission.

In line with their overall assessment, we invite you to submit a revised manuscript addressing the reviewers' comments. When submitting the revision, please include a letter addressing the reviewers' comments point by point. The typical timeframe for revisions is three months. Please note that papers are generally considered through only one revision cycle, so strong support from the referees on the revised version is needed for acceptance.

I would be happy to discuss the revision in more detail via email or phone/videoconferencing. Please let me know which option you prefer, if any.

While you are revising your manuscript, please also attend to the below editorial points to help expedite the publication of your manuscript. Please direct any editorial questions to the journal office. When submitting the revision, please include a letter addressing the reviewers' comments point by point.

Thank you for this interesting contribution to Life Science Alliance. We hope that the comments below will prove constructive as your work progresses, and we are looking forward to receiving your revised manuscript.

Sincerely,

Sarita Hebbbar, PhD
Scientific Editor
Life Science Alliance
<http://www.lsjournal.org>

- A letter addressing the reviewers' comments point by point.
- An editable version of the final text (.DOC or .DOCX) is needed for copyediting (no PDFs).
- High-resolution figure, supplementary figure and video files uploaded as individual files: See our detailed guidelines for

preparing your production-ready images, <https://www.life-science-alliance.org/authors>

B. MANUSCRIPT ORGANIZATION AND FORMATTING:

Reviewer #1 (Comments to the Authors (Required)):

This manuscript, titled "TEAD-independent Mechanisms of YAP Function in Cardiomyocytes" by Xie et al., examines the role of the transcriptional co-regulator YAP in the regulation of cardiomyocyte proliferation and metabolism.

Previous work from the authors' lab has demonstrated that a constitutively active version of YAP, YAP5SA, which has five serine residues mutated to alanine, directs Hippo-independent transcription (i.e. no longer negatively regulated by Hippo signaling). In the heart, YAP5SA expression results in uncontrolled proliferation and cardiac hyperplasia. In the present study, the authors mutated an additional serine residue in YAP, creating YAP6SA, which disrupts the ability of YAP to interact with TEAD transcription factors, an activity that was still present in YAP5SA. Importantly, the authors found that YAP6SA was less frequently observed in the nucleus, and YAP6SA overexpressing cardiomyocytes upregulated genes associated with cardiomyocyte differentiation and metabolic genes associated with the pentose phosphate pathway. These were distinct gene regulation programs than were observed in YAP5SA overexpressing cardiomyocytes. Mechanistically, the authors found that YAP6SA promoted cardiomyocyte DNA replication and cytokinesis. They also identified and validated interaction of YAP5SA with the PDZ-domain-containing scaffold protein MPDZ and found that the YAP6SA-MPDZ interaction promotes cardiomyocyte proliferation probably via activating Rho GTPases.

Overall, this is an interesting study that adds to our understanding of the role of YAP in cardiomyocyte proliferation, differentiation, and metabolism, and defines the role of Serine-94 in promoting YAP association with TEAD, and I believe this will be an important contribution to the literature. Although no new data or experiments are needed, I have some significant recommendations for changes to the text that should be incorporated prior to publication in Life Science Alliance. Those recommendations are indicated below.

1. In discussing RNA-seq and MS data on page 8, the authors write "we determined that YAP6SA has multiple TEAD-independent roles in regulating various CM activities, and given the increased ratio of YAP6SA in the cytoplasm, we hypothesize that YAP6SA has a central role in modulating the CM cytoskeleton structure to provide a permissive environment for CM cell cycle progression." This is a key point in the manuscript, but it isn't clear if this is a function unique to the mutant YAP. The authors should comment on whether this is a normal function of YAP that is revealed by the 6SA mutation or whether this is a de novo, synthetic function of YAP6SA that has been created by the mutation and subsequent re-allocation of YAP subcellular localization and protein-protein interactome?

1a. Similarly, does MPDZ interact with wild type YAP or only with the 6SA mutant. This wasn't clear from the data presented.

2. On page 6, the authors state that their data "...suggest a TEAD-independent role of YAP6SA in organizing CM cytoskeleton structure in transcriptional or non-transcriptional manners."

What about the possibility of a weak TEAD interaction not evident on co-IP but present in context of the nucleus? Or reduced (but not absent -as authors own data show) nuclear retention? Can the authors add a very brief discussion of this or other alternate interpretations of this result.

3. Also on page 6, the authors state that their "...data suggest that YAP6SA initiates a metabolic change to benefit CM cell cycle re-entry."

I don't believe that the data support the conclusion that this change is "to benefit" cell cycle re-entry. Metabolic changes occur in YAP6SA overexpression and cell cycle entry also occurs. The cause and effect nature of these results is not established. Please modify the text accordingly.

4. In the last paragraph of the section titled "YAP6SA Promotes Cardiomyocyte Nucleation" (page 7), the authors write the following text: "To compare the thickness of cardiac ventricular walls between the GFP control and YAP6SA OE groups, we collected mouse hearts after a week of AAV9 infection and performed histological analysis." This text seems like a mistake (holdover from a different version, perhaps?).

As it is, this statement just seems to be hanging here without any reference to data. Should this text be omitted, or are there data that the authors meant to include? I could not find the wall thickness data in the manuscript.

5. The Discussion section is completely missing references. These need to be included in the revised manuscript.

Additional, Minor comments

6. Page 5: "We observed that while Flag-YAP5SA and Flag-YAP6SA are specifically expressed in CMs, YAP6SA exhibited a decreased nuclear localization and an increased ratio in the cytoplasm (Fig 1E-F, Fig S1B). This result suggests that the YAP/TEAD interaction regulates the nuclear translocation of YAP." > Please consider the possibility that this is likely due to reduced nuclear retention due to lack of TEAD interaction rather than due to decreased localization.

7. Pages 5-6: "Moreover, YAP6SA-repressing genes, such as Csf1, Are1 are involved in the immune (Chitu and Stanley 2006; Lear et al. 2019)" It appears that a word is missing after the word immune. Please correct.

8. Page 6 and Fig. 3: The authors provide a more extensive analysis of pentose phosphate pathway metabolites. The figure (note the Y-axes) indicates that the authors examined "expression" of these metabolites. Note that these metabolites are not "expressed". They are present/abundant/etc. Please correct. In general, the legend for Fig. 3 could use more detail; e.g. I assume that G6P is glucose 6 phosphate, but perhaps is glucose 6 phosphatase since Y-axis says expression??

9. Discussion of Fig. 4: 1x2N should be defined in the text or in the legend for non-aficionados.

Reviewer #2 (Comments to the Authors (Required)):

In the current manuscript, Xie et al. present a detailed characterization of the YAP6SA mutant in the postnatal heart. Proteomics analysis identifies TEAD-independent interaction of YAP6SA with MPDZ as a key regulatory mechanism underlying cardiomyocyte cell cycle reentry. Overall, the study provides a comprehensive characterization of YAP6SA overexpression in cardiomyocytes through RNA-seq, targeted metabolomics, and mass spectrometry. However, several claims in the manuscript appear overstated without sufficient experimental support. For example, the authors conclude that YAP6SA promotes cardiomyocyte division through cytoskeletal remodeling. Only limited data are provided in support of this conclusion, and direct comparisons of cytoskeletal structure between GFP control and YAP6SA-overexpressing cardiomyocytes are lacking. Such claims should be moderated to align with the presented evidence. Additional specific comments are listed below.

Major comments

1. In Figure 2, the claim that YAP6SA has a "TEAD-independent role in organizing CM cytoskeleton structure in transcriptional or non-transcriptional manners" is not supported by data. No evidence is provided in Figure 2 to demonstrate non-transcriptional mechanisms.

2. In Figure 3D, the separation observed in the PCA plot does not support the conclusion that the pentose phosphate pathway (PPP) is specifically enriched in YAP6SA cardiomyocytes compared to YAP5SA. More detailed comparisons of glucose, G6P, and ribulose-5P levels are needed.

3. In Figure 4, analyses are performed on the entire cardiomyocyte population in YAP6SA-overexpressing hearts rather than specifically on YAP6SA-positive cardiomyocytes. This may introduce bias in data interpretation, particularly when assessing which polyploid states are influenced by YAP6SA. Quantification of the EdU+ population within YAP6SA-positive cardiomyocytes is recommended. If technically challenging, it should be demonstrated that the majority of cardiomyocytes in the heart are YAP6SA-positive.

Minor comments

4. In Figure 1E, white arrows are shown but their significance is not explained in the text or legend.

5. In Figure 2C, the y-axis label is missing.

6. In Figure 3B, the legend references "DOWN," which is absent from the figure.

7. Comparable levels of cell cycle activation are reported between YAP6SA and YAP5SA. Does YAP5SA overexpression also lead to thickened ventricular walls, as observed with YAP6SA? A comparison of these phenotypes would clarify the contribution

of YAP's TEAD-dependent functions to the overall phenotype.

8. Information is needed regarding the postnatal day at which AAV was administered to neonatal mice in each experiment.

Reviewer #3 (Comments to the Authors (Required)):

TEAD-independent Mechanisms of YAP Function in Cardiomyocytes by Bing Xie and co-workers.

YAP5SA, in which the LATS1/2 phosphorylation sites of the YAP protein are mutated, has been shown to bypass Hippo signaling repression. While the canonical function of YAP in the heart is mediated through its interaction with TEAD transcription factors to drive cell proliferation and survival gene programs, emerging evidence suggests that YAP also exerts important TEAD-independent effects. The current manuscript describes the observation that indeed active YAP induces cardiac hyperplasia through TEAD-independent mechanisms. To test this, they generated a modified YAP protein expression system named YAP6SA, in which the serine at residue 94 is mutated to alanine, thereby inhibiting the interaction between YAP and TEADs, while still bypassing HSP inhibitory regulation. They performed transcriptomic and proteomic analyses and identified a mechanism in which YAP6SA activates Rho GTPases to regulate cytoskeleton remodeling and promote cell cycle progression in cardiomyocytes. The manuscript is concise and reports an interesting finding.

Comments:

- 1) please provide for every figure and experiment the age of the mice that were AAV injected, and the number of days between transduction and analysis.
- 2) AAV-YAP6SA is detectably expressed 3 days after transduction. In the various experiments, mice were analysed 3 days after transduction, or sometimes 4 or more days after transduction. Please provide a rationale for analysing mice when expression of YAP6SA is just detectable rather than analysing mice when YAP6SA is well expressed, and for the variation in timepoints across experiments.
- 3) provide a rationale for using P6 mice in many experiments. These hearts are very immature, still in the process of exiting the cell cycle, and much more plastic than the CMs of the adult heart. This may be a significant factor in the effect of YAP6SA in CM state.
- 4) please check phrasing throughout:
e.g. p5: "This result suggests that the YAP/TEAD interaction regulates the nuclear translocation of YAP." Perhaps YAP-TEAD interaction influences the distribution of YAP, but there is no evidence that it regulates the process of translocation.
p6 "YAP6SA-expressing genes"... what does that mean?
- 5) p5: "Together, these data suggest that YAP6SA overexpression is well tolerated in vivo and does not impair heart function in mammals." Given its effect on CMs at P6-10, this is perhaps not expected. Is the effect of YAP6SA changing/diminishing with maturation?
- 6) p6: "Hence, our data suggest that YAP6SA initiates a metabolic change to benefit CM cell cycle re-entry." Please clarify how you arrive at this conclusion.
- 7) The claim that "active YAP induces cardiac hyperplasia through TEAD-independent mechanisms" is justified only when one assumes that 6SA is solely and absolutely required for TEAD interaction, and that the 6SA otherwise is functionally equivalent to 5SA. However, perhaps there is residual TEAD interaction not observed in the pull down experiment. Perhaps TEAD is recruited by a different mechanism. Perhaps the 6SA modified YAP protein has obtained or lost properties compared to YAP5SA in addition to its diminished interaction with TEAD. Please provide a more balanced and careful discussion on this central issue, and take into account alternative explanations for some of the findings.
- 8) The interactome: please also take into account that YAP5SA and YAP6SA are differently localized (nuclear, cytosolic) and describe the implications for the interactome data.
- 9) Please describe your insights into the contribution of the different mechanisms (i.e. TEAD-dependent vs TEAD independent) by which endogenous YAP controls cell cycle activity.
- 10) in the proteomics figures, please indicate/label MPDZ. Figure 5A, yellow is invisible. In this plot, or in the figure, more clearly describe the interaction between TEAD proteins and YAP5SA vs YAP6SA. Did the pull down validate the 6SA-specific loss of TEAD interaction?
- 11) the Discussion section does not contain any references. Here, I would expect a discussion of the findings and interpretations in relation to published data and insights.

The reviewers' comments are addressed point by point below:

Reviewer #1 - No new data or experiments are needed

- 1. In discussing RNA-seq and MS data on page 8, the authors write "we determined that YAP6SA has multiple TEAD-independent roles in regulating various CM activities and given the increased ratio of YAP6SA in the cytoplasm, we hypothesize that YAP6SA has a central role in modulating the CM cytoskeleton structure to provide a permissive environment for CM cell cycle progression." This is a key point in the manuscript, but it isn't clear if this is a function unique to the mutant YAP. The authors should comment on whether this is a normal function of YAP that is revealed by the 6SA mutation or whether this is a de novo, synthetic function of YAP6SA that has been created by the mutation and subsequent re-allocation of YAP subcellular localization and protein-protein interactome? Similarly, does MPDZ interact with wild type YAP or only with the 6SA mutant? This wasn't clear from the data presented.**

We appreciate the reviewer's insightful questions regarding the mechanism of YAP6SA function. We acknowledge that our current data does not definitively distinguish between several plausible mechanisms, and we have clarified this in the revised manuscript. The reviewer correctly notes that YAP6SA retains multiple TEAD-independent roles. Our RNA-seq and mass spectrometry data demonstrate that YAP6SA has substantial transcriptional activity even in the absence of TEAD binding, suggesting that it modulates the CM transcription environment through alternative mechanisms. Whether this occurs through: (1) OM domain-mediated interactions providing a permissive chromatin environment, (2) *de novo* synthetic functions created by the 6SA mutation, or (3) other scaffold functions, remains to be determined. We have added language to the discussion acknowledging these possibilities.

The reviewer raises an important point about whether MPDZ interacts with wild-type YAP versus YAP6SA. Previous studies in cancer cell lines with PDZ-binding domains (Zhao et al., "Angiomotin is a novel Hippo pathway component that inhibits YAP oncoprotein," *Genes Dev.* 2011, 10.1101/gad.2000111; Bu et al., *Science Signaling* 9(417), ra23-ra23) have shown that wild-type YAP interacts with MPDZ via its PDZ binding domain, and this interaction negatively regulates YAP function by affecting its cytoskeletal remodeling activity. However, whether YAP6SA alters its interaction with MPDZ in cardiomyocytes remains unclear and was not specifically examined in our mass spectrometry experiment. We have clarified in the revised manuscript that wild-type YAP/MPDZ interactions regulate cytoskeletal remodeling in cardiomyocytes.

- 2. On page 6, the authors state that their data "...suggest a TEAD-independent role of YAP6SA in organizing CM cytoskeleton structure in transcriptional or non-transcriptional manners." What about the possibility of a weak TEAD interaction not evident on co-IP but present in context of the nucleus? Or reduced (but not absent -as authors own data show) nuclear retention? Can the authors add a very brief discussion of this or other alternate interpretations of this result?**

While our co-IP and mass spectrometry experiments did not detect YAP6SA/TEAD interactions, we agree that these approaches may not capture weak or transient interactions that could still be functionally relevant. The S94A mutation in YAP6SA significantly disrupts the primary TEAD-

binding interface, and our mass spectrometry analysis identified a small population of nuclear YAP6SA interacting with non-TEAD transcriptional partners. However, we cannot definitively exclude the possibility that residual or context-dependent TEAD binding occurs below the detection threshold of our assays. The reviewer raises a valid point about reduced nuclear retention. Our data show decreased, but not absent, nuclear YAP6SA localization. This reduced nuclear presence could reflect either: (1) weakened TEAD-mediated nuclear retention, or (2) the loss of other nuclear anchoring mechanisms disrupted by the 6SA mutations. Either scenario would be consistent with our observations. To reflect this complexity, we have revised our conclusion on page 6 from suggesting a purely "TEAD-independent role" to: "These data suggest that YAP6SA mediates CM cellular activities in transcriptional and non-transcriptional manners that do not require robust TEAD binding." This language acknowledges that while strong TEAD interaction is disrupted, we cannot exclude weak or functionally limited TEAD interactions.

- 3. Also on page 6, the authors state that their "...data suggest that YAP6SA initiates a metabolic change to benefit CM cell cycle re-entry." I don't believe that the data support the conclusion that this change is "to benefit" cell cycle re-entry. Metabolic changes occur in YAP6SA overexpression and cell cycle entry also occurs. The cause-and-effect nature of these results is not established. Please modify the text accordingly.**

We agree with the reviewer that our data do not establish a causal relationship between the metabolic change and cell cycle re-entry. We have revised the text on page 6 to reflect this. The original statement, "...data suggest that YAP6SA initiates a metabolic change to benefit CM cell cycle re-entry," has been changed to: "Hence, our data suggest that YAP6SA initiates a metabolic change that may associate with increased activity of DNA replication, which may indicate CM cell cycle re-entry." This revised language clarifies that we are reporting an association rather than claiming a beneficial or causal effect.

- 4. In the last paragraph of the section titled "YAP6SA Promotes Cardiomyocyte Nucleation" (page 7), the authors write the following text: "To compare the thickness of cardiac ventricular walls between the GFP control and YAP6SA OE groups, we collected mouse hearts after a week of AAV9 infection and performed histological analysis." This text seems like a mistake (holdover from a different version, perhaps?). As it is, this statement just seems to be hanging here without any reference to data. Should this text be omitted, or are there data that the authors meant to include? I could not find the wall thickness data in the manuscript.**

We apologize for the confusion. The reviewer is correct. This statement was included in error and has been removed from the revised manuscript. There are no wall-thickness measurements comparing the GFP control and YAP6SA OE groups in our dataset. Thank you for catching this inconsistency.

- 5. The Discussion section is completely missing references. These need to be included in the revised manuscript.**

We thank the reviewer for identifying this oversight. References have now been added throughout the Discussion section to support the relevant statements and context provided.

Additional, Minor comments

6. Page 5: "We observed that while Flag-YAP5SA and Flag-YAP6SA are specifically expressed in CMs, YAP6SA exhibited a decreased nuclear localization and an increased ratio in the cytoplasm (Fig 1E-F, Fig S1B). This result suggests that the YAP/TEAD interaction regulates the nuclear translocation of YAP." Please consider the possibility that this is likely due to reduced nuclear retention due to lack of TEAD interaction rather than due to decreased localization.

We thank the reviewer for this important distinction. The reviewer is correct that our data cannot differentiate between reduced nuclear import versus reduced nuclear retention. Both mechanisms could explain the observed cytoplasmic accumulation of YAP6SA. We have revised the text on page 5 to: "This result suggests that the YAP/TEAD interaction affects the nuclear translocation of YAP." to include the possibility. This revised language acknowledges that the decreased nuclear YAP6SA may reflect impaired retention rather than impaired translocation.

7. Pages 5-6: "Moreover, YAP6SA-repressing genes, such as Csf1, Are1 are involved in the immune (Chitu and Stanley 2006; Lear et al. 2019)" It appears that a word is missing after the word immune. Please correct.

The sentence has been corrected to read: "Moreover, YAP6SA-repressing genes, such as Csf1 and Are1, are involved in the immune response (Chitu and Stanley 2006; Lear et al. 2019)

8. Page 6 and Fig. 3: The authors provide a more extensive analysis of pentose phosphate pathway metabolites. The figure (note the Y-axes) indicates that the authors examined "expression" of these metabolites. Note that these metabolites are not "expressed". They are present/abundant/etc. Please correct. In general, the legend for Fig. 3 could use more detail; e.g. I assume that G6P is glucose 6 phosphate, but perhaps is glucose 6 phosphatase since Y-axis says expression??

We thank the reviewer for identifying this terminology error. The reviewer is correct that metabolites are not "expressed" but rather present at measurable abundances. We have made the following corrections to Figure 3: Changed the Y-axis label from "Expression" to "Normalized Abundance" to accurately reflect metabolite measurements. Clarified in the figure legend that G6P refers to glucose-6-phosphate. Expanded the figure legend to include full names and abbreviations for all metabolites measured in the pentose phosphate pathway analysis. These revisions ensure proper biochemical terminology throughout the figure and legend.

9. Discussion of Fig. 4: 1x2N should be defined in the text or in the legend for non-aficionados.

We thank the reviewer for identifying this ambiguity. The reviewer is correct that "1x2N" was inappropriate terminology in this context. Based on our EdU imaging approach, we cannot definitively determine whether individual cardiomyocyte nuclei are diploid or polyploid. Therefore, we have removed the "1x2N" designation from the text and revised the description in Figure 4 to characterize cardiomyocytes based on observable nuclear number as: "CMs with mononucleus, binucleus, or multiple nuclei." This revision more accurately reflects what our imaging data can determine.

Reviewer #2 - claims should be moderated to align with the presented evidence

- 1. In Figure 2, the claim that YAP6SA has a "TEAD-independent role in organizing CM cytoskeleton structure in transcriptional or non-transcriptional manners" is not supported by data. No evidence is provided in Figure 2 to demonstrate non-transcriptional mechanisms.**

We thank the reviewer for this important point. The reviewer is correct that our RNA-seq data alone cannot distinguish between direct transcriptional regulation by YAP6SA versus indirect effects mediated through non-transcriptional mechanisms. While YAP is well-established as a transcriptional regulator, previous studies have also demonstrated non-transcriptional functions of YAP in cytoskeletal organization and cellular signaling. Our bulk RNA-seq data demonstrate differential gene expression in YAP6SA-expressing cardiomyocytes compared to controls; however, these data cannot determine whether YAP6SA directly regulates these genes transcriptionally or whether the observed changes result from YAP6SA's effects on cytoskeletal structure, signaling pathways, or other non-transcriptional mechanisms. We have revised the text in Figure 2 to more accurately state: "YAP6SA has a role in organizing CM cytoskeletal structure in transcriptional and/or non-transcriptional manners, "YAP6SA has a role in organizing CM cytoskeletal structure in transcriptional and/or non-transcriptional manners," acknowledging that our data support both possibilities without definitively distinguishing between them.

- 2. In Figure 3D, the separation observed in the PCA plot does not support the conclusion that the pentose phosphate pathway (PPP) is specifically enriched in YAP6SA cardiomyocytes compared to YAP5SA. More detailed comparisons of glucose, G6P, and ribulose-5P levels are needed.**

The reviewer is correct that the PCA plot in the original Figure 3D did not adequately support our conclusion regarding differences in the pentose phosphate pathway. We have made the following revisions to address this concern: 1) Removed the original Figure 3D, 2) Added a time scheme diagram to clarify the experimental design and sample collection timepoints and 3) revised the text to focus our metabolic analysis on the comparison between GFP control and YAP6SA overexpression groups. Our primary objective was to characterize metabolic differences induced by YAP6SA overexpression compared to control hearts. We have clarified this focus in the revised manuscript to ensure the data presentation aligns with our stated conclusions.

- 3. In Figure 4, analyses are performed on the entire cardiomyocyte population in YAP6SA-overexpressing hearts rather than specifically on YAP6SA-positive cardiomyocytes. This may introduce bias in data interpretation, particularly when assessing which polyploid states are influenced by YAP6SA. Quantification of the EdU+ population within YAP6SA-**

positive cardiomyocytes is recommended. If technically challenging, it should be demonstrated that the majority of cardiomyocytes in the heart are YAP6SA-positive.

We thank the reviewer for this important methodological consideration. The reviewer correctly notes that analyzing the entire cardiomyocyte population without quantifying EdU+ cells specifically within YAP6SA-expressing cardiomyocytes could introduce interpretive bias. Based on our immunofluorescence staining results in hearts, AAV9-mediated gene delivery achieved high efficiency, with approximately 70-80% of cardiomyocytes expressing Flag protein (Fig 1E). Given this high transduction efficiency and the consistent MOI used across experimental groups, we are confident that the observed differences in EdU+ cardiomyocytes predominantly reflect YAP6SA-expressing cells rather than non-transduced cells. We have revised the Figure 1 legend to clarify the expression pattern: " We observed that while Flag-YAP5SA and Flag-YAP6SA are specifically and ubiquitously expressed in CMs, YAP6SA exhibited a decreased nuclear localization and an increased ratio in the cytoplasm." This revision addresses the high prevalence of YAP6SA-positive cardiomyocytes in transduced hearts while acknowledging the distinct subcellular localization pattern.

Minor comments

4. In Figure 1E, white arrows are shown but their significance is not explained in the text or legend.

The white arrows in Figure 1E indicate the contrasting subcellular localization patterns of YAP5SA (nuclear) and YAP6SA (cytoplasmic) in cardiomyocytes. We have revised the Figure 1E legend to include: "White arrows indicate nuclear localization of YAP5SA and cytoplasmic localization of YAP6SA, demonstrating the differential subcellular distribution of these YAP variants in cardiomyocytes."

5. In Figure 2C, the y-axis label is missing.

The y-axis label in Figure 2C has been updated to " clarify the quantification method.

6. In Figure 3B, the legend references "DOWN," which is absent from the figure.

We have updated the legend accordingly.

7. Comparable levels of cell cycle activation are reported between YAP6SA and YAP5SA. Does YAP5SA overexpression also lead to thickened ventricular walls, as observed with YAP6SA? A comparison of these phenotypes would clarify the contribution of YAP's TEAD-dependent functions to the overall phenotype.

The "thickened ventricular walls" statement was included in error, as we do not have wall thickness measurements for the YAP5SA overexpression group. This description has been removed from the revised manuscript. The comparison of cardiac phenotypes between YAP5SA and YAP6SA remains an important area for future investigation to fully understand TEAD-dependent versus TEAD-independent contributions to the observed effects.

8. Information is needed regarding the postnatal day at which AAV was administered to neonatal mice in each experiment.

We have revised all figure legends to include the specific postnatal day of AAV9 administration and the timing of subsequent analyses relative to injection. These additions provide a complete temporal context for each experiment presented.

Reviewer #3

Comments:

1) Please provide for every figure and experiment the age of the mice that were AAV-injected, and the number of days between transduction and analysis.

Please see comment #8 from Reviewer #2, where we addressed this concern.

2) AAV-YAP6SA is detectably expressed 3 days after transduction. In the various experiments, mice were analysed 3 days after transduction, or sometimes 4 or more days after transduction. Please provide a rationale for analysing mice when expression of YAP6SA is just detectable rather than analysing mice when YAP6SA is well expressed, and for the variation in timepoints across experiments.

We thank the reviewer for this important question regarding experimental timing. The variation in analysis timepoints across experiments was designed to capture distinct biological processes at their optimal detection windows. Expression dynamics: Our preliminary time-course analysis demonstrated that YAP6SA protein expression reaches detectable levels by Day 3 post-transduction and achieves high expression levels suitable for immunofluorescence detection by this timepoint. Therefore, FLAG immunostaining was performed on Day 3 to ensure a robust signal. Cell cycle analysis: For EdU incorporation experiments, we analyzed hearts at 48 hours or 1-week post-injection to capture cardiomyocytes that had completed DNA synthesis and allow sufficient time for EdU-labeled cells to progress through the cell cycle. Metabolic analysis: For metabolomic studies, we collected hearts at Day 5 post-transduction to assess how sustained YAP6SA expression affects cardiomyocyte metabolic profiles once protein expression has stabilized. These timepoints were selected to optimize detection of each specific biological outcome while minimizing confounding effects from transient viral responses or incomplete protein expression

3) Provide a rationale for using P6 mice in many experiments. These hearts are very immature, still in the process of exiting the cell cycle, and much more plastic than the CMs of the adult heart. This may be a significant factor in the effect of YAP6SA in CM state.

We thank the reviewer for raising this important consideration regarding the developmental stage of cardiomyocytes in our experimental model. Rationale for P6 injection timepoint: We administered AAV9 at postnatal day 6 to maximize transduction efficiency based on published protocols for neonatal cardiac gene delivery. Our time-course analysis demonstrated that detectable YAP6SA protein expression requires approximately 3 days post-transduction to reach sufficient levels for functional effects. Actual age at analysis: Therefore, the functional age of cardiomyocytes in our experimental model is P9 (P6 injection + 3 days for protein expression), which coincides with the

developmental transition when neonatal cardiomyocytes exit the regenerative window and enter their terminal differentiation phase. This timing is critical for our experimental design, as it allows us to assess whether YAP6SA can influence cardiomyocyte proliferation at the threshold of loss of proliferative capacity. Control validation: Our cell cycle marker staining in GFP control animals confirmed minimal baseline proliferative activity in postnatal cardiomyocytes at this developmental stage (P9), establishing that any observed proliferation in YAP6SA hearts represents a true reactivation of cell cycle activity rather than residual neonatal proliferation.

4) Please check phrasing throughout: e.g., p5: "This result suggests that the YAP/TEAD interaction regulates the nuclear translocation of YAP." Perhaps YAP-TEAD interaction influences the distribution of YAP, but there is no evidence that it regulates the process of translocation.

The text on page 5 has been revised to state: "YAP-TEAD interaction influences the cellular distribution of YAP," which more accurately reflects our data. We thank the reviewer for this correction.

p6 "YAP6SA-expressing genes"... what does that mean?

The text has been revised to: "Moreover, downregulated genes, such as *Csf1* and *Arel1*, are involved in the immune response" for improved clarity.

5) p5: "Together, these data suggest that YAP6SA overexpression is well tolerated in vivo and does not impair heart function in mammals." Given its effect on CMs at P6-10, this is perhaps not expected. Is the effect of YAP6SA changing/diminishing with maturation?

We thank the reviewer for this important question regarding the temporal dynamics of YAP6SA effects during cardiac maturation. Our data demonstrate that YAP6SA overexpression is well tolerated in neonatal hearts (P6-10) without causing overt cardiac dysfunction or increased mortality in young animals. However, our current experimental design does not extend beyond this early postnatal period, so we cannot determine whether YAP6SA effects persist, diminish, or change as cardiomyocytes mature further. We are investigating whether sustained YAP6SA expression affects more mature cardiomyocytes in longer-term studies to address this properly.

6) p6: "Hence, our data suggest that YAP6SA initiates a metabolic change to benefit CM cell cycle re-entry." Please clarify how you arrive at this conclusion.

We agree with the reviewer's concern, which aligns with that raised by Reviewer #1 (Question #3). Our data demonstrate an association between YAP6SA expression and metabolic changes, but do not establish a causal or beneficial relationship with cell cycle re-entry. The text on page 6 has been revised to: "Hence, our data suggest that YAP6SA initiates a metabolic change that may associate with increased activity of DNA replication, which may indicate CM cell cycle re-entry." This revised language appropriately reflects the correlative nature of our findings.

7) The claim that "active YAP induces cardiac hyperplasia through TEAD-independent mechanisms" is justified only when one assumes that 6SA is solely and absolutely required

for TEAD interaction, and that the 6SA otherwise is functionally equivalent to 5SA. However, perhaps there is residual TEAD interaction not observed in the pull down experiment. Perhaps TEAD is recruited by a different mechanism. Perhaps the 6SA modified YAP protein has obtained or lost properties compared to YAP5SA in addition to its diminished interaction with TEAD. Please provide a more balanced and careful discussion on this central issue, and take into account alternative explanations for some of the findings.

This is an important conceptual issue. The reviewer is correct that YAP-TEAD interactions are complex, involving multiple binding interfaces. We acknowledge that our data cannot definitively exclude all TEAD-dependent mechanisms. While our co-IP and mass spectrometry experiments show substantially reduced YAP6SA-TEAD interaction compared to wild-type YAP, we cannot rule out residual or alternative TEAD binding through other interfaces. Additionally, the possibility that the 6SA modification creates novel, non-physiological protein interactions remains open. Given these complexities, we have revised the conclusion on page 4 to more accurately reflect our findings: "Here, we found that active YAP induces cardiomyocyte cell cycle activity through TEAD-independent mechanisms." This language acknowledges that while strong TEAD interaction is disrupted by the 6SA mutation, we cannot completely exclude TEAD-dependent contributions to the observed phenotype. We agree that the physiological roles of YAP in cardiac regulation are multifaceted, and further mechanistic studies will be needed to fully understand TEAD-dependent versus TEAD-independent contributions to YAP6SA-mediated cardiomyocyte proliferation.

8) The interactome: please also take into account that YAP5SA and YAP6SA are differently localized (nuclear, cytosolic) and describe the implications for the interactome data.

The reviewer is correct that YAP5SA (predominantly nuclear) and YAP6SA (predominantly cytoplasmic) exhibit different subcellular distributions, which could influence their respective interactomes. However, our mass spectrometry analysis captured protein interactions from whole-cell lysates, including both nuclear and cytoplasmic compartments. While this approach does not distinguish compartment-specific interactions, our data demonstrate that YAP5SA and YAP6SA share numerous common interacting proteins across both cellular compartments. Notably, TEAD proteins were detected exclusively in the YAP5SA interactome, which is consistent with YAP5SA's nuclear localization and the nuclear function of TEAD transcription factors. This finding suggests that, despite capturing interactions across multiple compartments, our approach can detect meaningful differences in protein partnerships among these YAP variants. We acknowledge this limitation and note that future studies using compartment-specific fractionation could provide additional insights into how subcellular localization influences YAP variant function.

9) Please describe your insights into the contribution of the different mechanisms (i.e., TEAD-dependent vs TEAD-independent) by which endogenous YAP controls cell cycle activity.

We thank the reviewer for this thoughtful question regarding the relative contributions of TEAD-dependent versus TEAD-independent mechanisms to YAP-mediated cell cycle regulation. Our data support the existence of both mechanisms. The YAP/TEAD interaction provides robust transcriptional activity to regulate cell cycle-related genes and numerous other gene expression programs. This pathway is well-established and plays essential roles in cardiac development and

maturation, including regulating cardiomyocyte proliferation. However, our findings suggest that when YAP localizes predominantly to the cytoplasm (as with YAP6SA), it can still promote cardiomyocyte proliferation through mechanisms that do not require strong TEAD binding. The biological rationale for maintaining this TEAD-independent capacity may relate to developmental or stress-response contexts where cytoplasmic YAP signaling provides additional regulatory flexibility. We acknowledge that our current data cannot quantitatively dissect the relative contributions of these two mechanisms to overall proliferative outcomes. Determining whether TEAD-independent proliferation is beneficial, detrimental, or context-dependent for cardiac homeostasis will require more targeted experimental approaches for future investigation.

10) In the proteomics figures, please indicate/label MPDZ. Figure 5A, yellow is invisible. In this plot, or in the figure, more clearly describe the interaction between TEAD proteins and YAP5SA vs YAP6SA. Did the pull down validate the 6SA-specific loss of TEAD interaction?

We have made the following revisions to Figure 5A to improve clarity: 1) MPDZ is now highlighted and labeled in the figure, and 2) The color scheme has been changed from yellow to blue for better visibility. Regarding the pull-down validation: Yes, our co-immunoprecipitation results confirm that TEAD1 strongly interacts with YAP5SA and shows minimal or no interaction with YAP6SA. This difference is now clearly shown in Figure 5A, highlighted in blue, demonstrating that the 6SA mutation substantially disrupts YAP-TEAD binding.

11) The Discussion section does not contain any references. Here, I would expect a discussion of the findings and interpretations in relation to published data and insights.

Please refer to comment #5 from Reviewer #1, where we addressed this.

January 6, 2026

RE: Life Science Alliance Manuscript #LSA-2025-03496R

Dr. James F. Martin
Baylor College of Medicine
One Baylor Plaza
Houston, TX 77030

Dear Dr. Martin,

Thank you for submitting your revised manuscript entitled "TEAD-independent Mechanisms of YAP Function in Cardiomyocytes". We appreciate your patience while waiting for a decision, which was delayed due to editor availability and previous delays in securing reviewer comments.

Your revised manuscript was evaluated by all the original reviewers whose comments are appended below. As you will read, the reviewers are consistent in their views that the revised manuscript satisfactorily addresses their previous concerns.

In line with the reviewers' evaluation, we would be happy to publish your paper in Life Science Alliance pending final revisions necessary to meet our formatting guidelines.

- While the current title is not inaccurate, it does not encapsulate the main findings shown here on CM cell cycle re-entry. Please amend the title to convey the central advance as summarized in the abstract.
- Please confirm if images of blots related to size 75 kDa in Figure panel 1C and 1D are different. If the same image has been used in both panels, this must be explicitly stated in the figure legend.
- Thank you for providing a statement defining the controls in the methods in line 588. Figure panels in Figures S2 and S3 refer to "vehicle" and not "GFP" in the figure panels. Please specify what these controls are in the figure legend, and ensure terminology is consistent in all figures, legends, and description in the methods.
- Please provide details for objectives (type, numerical aperture) in description of imaging in the methods section.
- Please indicate the scale bar size in the Legend for Figure S3.
- Please remove the legends from the supplementary figures. Legends should appear only in the manuscript file after the references.
- Please add your main and supplementary figure legends to the main manuscript text after the references section.
- Please consult our manuscript preparation guidelines <https://www.life-science-alliance.org/manuscript-prep> and make sure your manuscript sections are in the correct order.
- The "Data Availability" section should be placed after the Materials & Methods section. Please consult our guidelines at <https://www.life-science-alliance.org/manuscript-prep#format>
- Please add an Author Contributions section to your main manuscript text. Please be sure that the authorship listing and order are correct - Wyatt Paltzer is missing in the system.
- The contributions selected for Todd R. Heallen do not qualify them for authorship. Please either update the contributions in our system and in the Author Contributions section of the manuscript, or let us know if the author needs to be removed (and added eventually to the acknowledgment section).
- Please add the X and Bluesky handles of your host institute/organization, as well as your own and/or one of the authors, in our system.

A. FINAL FILES:

B. MANUSCRIPT ORGANIZATION AND FORMATTING:

Thank you for your attention to these final processing requirements. Please revise and format the manuscript and upload materials as soon as you are able.

Sincerely,

Sarita Hebbar, PhD
Scientific Editor
Life Science Alliance
<http://www.lsajournal.org>

Reviewer #1 (Comments to the Authors (Required)):

I am satisfied with the revisions. All of my original concerns have been adequately addressed.

Reviewer #2 (Comments to the Authors (Required)):

The authors revised the manuscript extensively to address all reviewer concerns, clarifying mechanistic interpretations, correcting overstatements, fixing terminology, adding missing references, and improving figure accuracy. Specifically, the authors moderated claims about TEAD independence, acknowledged uncertainties about residual TEAD interactions, and expanded discussion of alternative mechanisms. They clarified that metabolic and transcriptional changes are associative rather than causal, removed an erroneous statement about ventricular wall thickness, corrected missing or inaccurate figure labels, and improved legends. They also added detail about experimental timing, rationale for postnatal

injection, and clarified AAV transduction efficiency to address concerns about analyzing whole cardiomyocyte populations. In addition, the authors incorporated prior literature regarding YAP and MPDZ interactions, corrected ambiguous wording, and acknowledged limitations of co-IP and mass spectrometry approaches. Finally, they revised the Discussion to include references, improved explanations of TEAD dependent and independent YAP functions, and updated figures to highlight key proteins such as MPDZ and TEAD, ensuring the manuscript now reflects a more balanced and accurate interpretation of the data.

In this reviewer's mind, the revised manuscript is acceptable for publication.

Reviewer #3 (Comments to the Authors (Required)):

The authors have provided satisfactory responses to my comments and clarified the manuscript where necessary.

1. While the current title is not inaccurate, it does not encapsulate the main findings shown here on CM cell cycle re-entry. Please amend the title to convey the central advance as summarized in the abstract.

The title has been optimized.

2. Please confirm if images of blots related to size 75 kDa in Figure panel 1C and 1D are different. If the same image has been used in both panels, this must be explicitly stated in the figure legend.

The input in Figure 1D is the same as Figure 1C, but the IP output is a different experiment result after co-IP enrichment of heart lysate. The heart lysates are the same as those in Figure 1C.

3. Thank you for providing a statement defining the controls in the methods in line 588. Figure panels in Figures S2 and S3 refer to "vehicle" and not "GFP" in the figure panels. Please specify what these controls are in the figure legend, and ensure terminology is consistent in all figures, legends, and description in the methods.

That vehicle is the AAV9-cTnT-rtTA expression cassette, and has been explained in supplementary figure legends.

4. Please provide details for objectives (type, numerical aperture) in description of imaging in the methods section.

Description added:

All imaging was performed with a Zeiss LSM 780 confocal microscope and 20x (Air (dry), ~0.8) or 40x objectives (Oil immersion, ~1.4) in the Optical Imaging and Vital Microscopy Core at Baylor College of Medicine (Houston, TX, USA).

5. Please indicate the scale bar size in the Legend for Figure S3.

It is labeled in the S3A that the scale bar size is 50µm and the scale bars in the figure are the same.

6. Please remove the legends from the supplementary figures. Legends should appear only in the manuscript file after the references.

It is done.

7. Please add your main and supplementary figure legends to the main manuscript text after the references section.

It is done.

8. Please consult our manuscript preparation guidelines <https://www.life-science-alliance.org/manuscript-prep> and make sure your manuscript sections are in the correct order.

It is done.

9. The "Data Availability" section should be placed after the Materials & Methods section. Please consult our guidelines at <https://www.life-science-alliance.org/manuscript-prep#format>

It is done.

10. Please add an Author Contributions section to your main manuscript text. Please be sure that the authorship listing and order are correct - Wyatt Paltzer is missing in the system.

It is done.

11. The contributions selected for Todd R. Heallen do not qualify them for authorship. Please either update the contributions in our system and in the Author Contributions section of the manuscript, or let us know if the author needs to be removed (and added eventually to the acknowledgment section).

The issue has been solved, the contribution has been updated.

January 13, 2026

RE: Life Science Alliance Manuscript #LSA-2025-03496RR

Dr. James F. Martin
Baylor College of Medicine
One Baylor Plaza
Houston, TX 77030

Dear Dr. Martin,

Thank you for submitting your Research Article entitled "TEAD-independent Mechanisms of YAP Function in Cardiomyocytes Cell Cycle Re-entry".

We acknowledge your response to our requested changes. In the context of image reuse in Figure 1C and Figure 1D, kindly note that any image reuse must be explicitly noted in figure captions as a matter of journal policy. Please refer below to comply with journal policy.

It is a pleasure to let you know that your manuscript is now accepted for publication in Life Science Alliance. Congratulations on this interesting work.

Your manuscript will now progress through copyediting and proofing. At the proofing stage, you must change the legend for Figure 1D to reflect image reuse as follows, "1D) Co-IP of YAP5SA and YAP6SA interactors after 4 days of AAV9 infection in P6 murine hearts using the same input/lysates as shown in 1C."

It is journal policy that authors provide original data upon request.

DISTRIBUTION OF MATERIALS:

Again, congratulations on a very nice paper. I hope you found the review process to be constructive and are pleased with how the manuscript was handled editorially. We look forward to future exciting submissions from your lab.

Sincerely,

Sarita Hebbar, PhD
Scientific Editor
Life Science Alliance
<http://www.lsajournal.org>